# CONFRONTING REWARD MODEL OVEROPTIMIZATION WITH CONSTRAINED RLHF

**Ted Moskovitz**[*]
Gatsby Unit, UCL

**Aaditya K. Singh**
Gatsby Unit, UCL

**DJ Strouse**
Google DeepMind

**Tuomas Sandholm**
Carnegie Mellon University[†]

**Ruslan Salakhutdinov**
Carnegie Mellon University

**Anca D. Dragan**
University of California, Berkeley

**Stephen McAleer**
Carnegie Mellon University

## ABSTRACT

Large language models are typically aligned with human preferences by optimizing *reward models* (RMs) fitted to human feedback. However, human preferences are multi-faceted, and it is increasingly common to derive reward from a composition of simpler reward models which each capture a different aspect of language quality. This itself presents a challenge, as it is difficult to appropriately weight these component RMs when combining them. Compounding this difficulty, because any RM is only a proxy for human evaluation, this process is vulnerable to *overoptimization*, wherein past a certain point, accumulating higher reward is associated with worse human ratings. In this paper, we perform, to our knowledge, the first study on overoptimization in composite RMs, showing that correlation between component RMs has a significant effect on the locations of these points. We then introduce an approach to solve this issue using constrained reinforcement learning as a means of preventing the agent from exceeding each RM's threshold of usefulness. Our method addresses the problem of weighting component RMs by learning dynamic weights, naturally expressed by Lagrange multipliers. As a result, each RM stays within the range at which it is an effective proxy, improving evaluation performance. Finally, we introduce an adaptive method using gradient-free optimization to identify and optimize towards these points during a single run.

## 1 INTRODUCTION

In the last several years, *Large Language Models* (LLMs) have made impressive advances in natural language processing. These models, which are typically pretrained on massive amounts of text data from the Internet to predict the next token given the current context, are often known as *foundation models* (Bommasani et al., 2021) for their ability to be adapted to a variety of downstream applications, such as chatbots (Brown et al., 2020; OpenAI, 2023; Touvron et al., 2023) or code generation (Ahmad et al., 2021; Wang et al., 2021; Rozière et al., 2023). This adaptation, or *finetuning*, is often performed via *reinforcement learning from human feedback* (RLHF; Knox and Stone, 2008; Christiano et al., 2017; Stiennon et al., 2020). RLHF treats the pretrained language model as a decision-making agent whose "actions" are tokens and whose goal is to maximize a *reward model* (RM) trained to emulate human preferences over output text. As these models become more prevalent in society, there are many concerns regarding their safe deployment (Hendrycks et al., 2023; Bubeck et al., 2023; Legg, 2008), including biases against marginalized or underrepresented groups (Bender et al., 2021), proliferation of false information (Lin et al., 2021), and leakage of sensitive information (Carlini et al., 2021). These concerns are collectively known as the *alignment problem*: how can we ensure that the behavior of these models is aligned with human preferences?

---

[*]Corresponding author: `ted@gatsby.ucl.ac.uk`
[†]Additional affiliations: Strategy Robot, Inc., Strategic Machine, Inc., Optimized Markets, Inc.

Current approaches to alignment within RLHF center around the collection of vast amounts of human rating data and the training of larger, more powerful RMs (Ouyang et al., 2022; Gao et al., 2022). However, a fundamental issue with any RM is that ultimately, it is only an imperfect proxy for human preferences. Gao et al. (2022) drew attention to this fact, showing that maximizing a reward model beyond a certain point can actually begin to decrease ground truth performance (*i.e.*, lead a text-based agent to produce outputs which are judged as qualitatively worse). This phenomenon is known as *reward model overoptimization*. Examples of overoptimization include producing overly wordy responses or hallucinating information in an effort to give the impression of expertise. One simple, yet expensive, approach to mitigating this issue is to periodically evaluate the model with fresh human rating throughout finetuning and stop early when ratings decline.

It is also increasingly common to derive reward from *composite RMs*: fixed combinations of several RMs each designed to capture a different aspect of text quality (Ramamurthy et al., 2022; Glaese et al., 2022; Yuan et al., 2023; Bakker et al., 2022; Wu et al., 2023). Such composite RMs are useful because they allow for more fine-grained measurement of agent behavior and each component can be retrained or swapped out without affecting the others. Despite these advantages, this approach also presents its own challenges. Determining the weighting among RMs requires hyperparameter optimization to find the combination that produces the best correlation with ground truth evaluation, and the risk of overoptimization means that the best weighting is contingent on a set training duration. Furthermore, when the reward is constructed from several RMs, information about each individual RM is lost, and the agent cannot attribute changes in reward to any single model. In particular, component rewards may even oppose one another, such as an RM which measures safety (and thus may deny certain user requests) versus another rewarding helpfulness (Bai et al., 2022). Worse, early stopping to avoid overoptimization in composite RMs is problematic, as different components will have different values at which they stop being effective proxies for human evaluation.

In this paper, we propose a simple approach to address these challenges: identify the points of overoptimization, which we term *proxy points*, and then use constrained optimization to ensure that each component RM reaches, but does not exceed, its associated proxy point. Rather than use a fixed weighting among components, our method dynamically adapts a weighting to modulate the influence of each RM on the learning process. The core idea behind our approach is to use these constraints to prevent the agent from overoptimizing its (composite) RM beyond the proxy points.

As in existing methods (Gao et al., 2022), we rely on some access to ground-truth queries. We propose two ways of using these queries to identify proxy points. In the first approach, we train multiple runs and track each reward model value, periodically querying the ground-truth reward model. This approach then finds an optimal joint proxy point by fitting a surface to this data and maximizing it. While effective, this approach requires multiple runs to fit the surface used to find proxy points. In the second approach, we speed up this process by only using one reinforcement learning run. As this run is training, we can periodically query the ground-truth reward model and use this data to run a derivative-free optimization algorithm to find the next candidate proxy points. To summarize, we make the following contributions:

- We provide analysis of reward model overoptimization in the context of composite reward functions, showing that the correlation between RMs has a significant influence on proxy points.

- We propose several constrained RL approaches which incorporate these points into the optimization objectives, preventing overoptimization and improving evaluation performance.

- We show that a derivative-free optimization method can be used to dynamically find these proxy points during a single run, significantly saving computation.

## 2 PRELIMINARIES: REINFORCEMENT LEARNING FROM HUMAN FEEDBACK

**RL Problem Formulation** In *reinforcement learning* (RL; Sutton and Barto, 2018), an agent seeks to take actions in its environment in order to maximize reward. Mathematically, this problem is typically formalized as a *Markov decision process* (MDP; Puterman, 2014), defined as a tuple $\mathcal{M} \triangleq (\mathcal{S}, \mathcal{A}, P, r, \gamma, \rho)$, where $\mathcal{S}$ is the state space, $\mathcal{A}$ is the action space, $P : \mathcal{S} \times \mathcal{A} \to \mathcal{P}(\mathcal{S})$ is the transition kernel (where $\mathcal{P}(X)$ denotes the set of distributions over $X$), $r : \mathcal{S} \times \mathcal{A} \times \mathcal{S} \to \mathbb{R}$ is the reward function, $\gamma \in [0, 1)$ is the discount factor, and $\rho \in \mathcal{P}(\mathcal{S})$ is the initial state distribution. In practice, the agent's experience is typically broken into discrete segments, or "episodes" of maximum length $T$. At the beginning of each episode, the environment resets and an initial state is sampled $s_0 \sim \rho(\cdot)$. At each time step $t = 0, 1, \ldots, T - 1$, the agent selects an action $a_t$ conditioned on

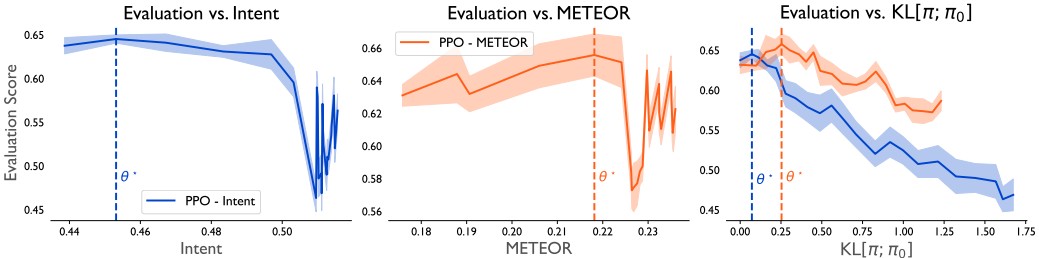

Figure 3.1: **Individual RMs are imperfect proxies for evaluation score.** Evaluation score initially increases as individual RMs and the KL divergence grow before falling at proxy points, denoted by dashed lines. Results are averaged over 5 seeds, with shading showing standard error.

its current state $s_t$ using a stationary policy $\pi(a_t|s_t)$, where $\pi : \mathcal{S} \to \mathcal{P}(\mathcal{A})$. Each episode can be summarized as a trajectory $\tau = (s_0, a_0, s_1, \ldots, s_T)$. The agent's goal is to find a policy with maximum expected *return* $R(\tau)$, where $R(\tau) \triangleq \sum_{t=0}^{T-1} \gamma^t r(s_t, a_t, s_{t+1})$. The expected return under policy $\pi$ is known as the *value* $v^\pi(s) \triangleq \mathbb{E}[R(\tau)|s_0 = s]$ or the *action-value* if conditioned on both states and actions $q^\pi(s, a) \triangleq \mathbb{E}[R(\tau)|s_0 = s, a_0 = a]$. The optimization problem faced by the agent, then, is $\max_\pi v^\pi$, where $v^\pi \triangleq \mathbb{E}_{s_0 \sim \rho(\cdot)} v^\pi(s_0)$ is the average value over initial states.

**Integrating Human Feedback**    The origin and nature of the reward is a fundamental question when formalizing a problem using RL. Using human evaluation to delineate good agent behaviors from bad has a history that extends beyond language models. Knox and Stone (2008) used human ratings of actions to construct a reward model for the game Tetris, while Christiano et al. (2017) proposed a mechanism for using human feedback to express preferences over trajectories collected in Atari and MuJoCo. In language modeling, each action is viewed as adding a new token to the current context string (Ziegler et al., 2019; Stiennon et al., 2020; Bai et al., 2022; Ouyang et al., 2022), which can be viewed as the state. The LM is then the policy, with action space $\mathcal{A}$ being the vocabulary of possible tokens, and state space $\mathcal{S}$ being the set of all sequences of tokens up to maximum length $T$. Transitions are deterministic, with each action token simply appended to the current state. Given a pretrained LM $\pi_0$, RLHF often consists of three stages (Casper et al., 2023): 1) collecting human feedback on model utterances (typically in the form of ranked preference data), 2) training a RM to model score utterances in alignment with human feedback (typically initialized from a separate pretrained LM) and 3) finetuning the LM with RL using the learned RM. While early work in RLHF for LLMs (Stiennon et al., 2020) focused on a single reward model, more recent work has shown performance benefits of using a weighted combination of simpler RMs (Wu et al., 2023).

**Overoptimization**    Recently, Gao et al. (2022) performed an empirical study of a phenomenon with deep ramifications for alignment: RM overoptimization. Their core finding is that after a certain point, increasing an LLM agent's value with respect to a given RM will actually begin to decrease its quality on the actual preferences it is trying to learn. (Gao et al. (2022) use a "gold standard" RM to stand in for human ratings for convenience.) The root of this issue is that any RM is only a proxy for the agent's true measuring stick—human evaluation—so as predicted by Goodhart's Law (Goodhart and Goodhart, 1984), an agent trained to maximize it will eventually learn behaviors which the true objective would discourage. Our approach to addressing this issue is based on a simple two-stage process: first, find the points where the available rewards stop being useful proxies, and second, train an agent to only maximize reward up until that point.

## 3 FINDING PROXY POINTS

**Setting**    In order to conduct an in-depth analysis given our available computational resources, we focus on a single setting as a case study: dialogue generation with the DailyDialog (Li et al., 2017) dataset, which consists of transcripts of conversations between humans. As input, the agent receives a snippet of conversation, and from this context, it must predict the next utterance. We describe this setting in detail in Appendix A. As a base LLM, we follow prior work (Wu et al., 2023) and use GPT-2 (Radford et al., 2019) here and throughout this paper. For the reward, we use a combination of two component rewards, each meant to capture a different element of desired behavior, to demonstrate our approach most directly. The first, $r^{met}$, is the METEOR score (Banerjee and Lavie, 2005) between

the generated utterance and reference output, which is computed based on a number of features, including word-matching, synonym-matching, and phrasing. The second, $r^{int}$, measures how well the intent of the generated utterance matches that of the reference output. It is computed using a fine-tuned RoBERTa model (Liu et al., 2019) which classifies text into different "intent categories" such as 'inform,' 'question,' or 'direct.' The typical approach (Ramamurthy et al., 2022) is to linearly combine these RMs to form a composite reward:

$$\tilde{r}_t = \alpha^{met} r_t^{met} + \alpha^{int} r_t^{int}, \tag{3.1}$$

where the coefficients $(\alpha^{met}, \alpha^{int})$ are fixed. As is standard in RLHF applied to language models, an additional KL penalty was added to discourage deviation from the initial model $\pi_0$:

$$r_t = \tilde{r}_t - \alpha_t^{\mathsf{KL}} \log \frac{\pi(a_t|s_t)}{\pi_0(a_t|s_t)}. \tag{3.2}$$

The coefficient $\alpha_t^{\mathsf{KL}}$ effectively acts as a Lagrange multiplier, increasing if the KL exceeds some threshold and decreasing otherwise. We discuss this in more detail in Appendix B.

**Evaluation and Proxy Points** In an ideal world, evaluation performance for all agents across all runs could be measured by collecting a large number of human ratings. However, this is expensive, so we instead selected a number of metrics other than METEOR and intent score which measure the lexical quality and di-

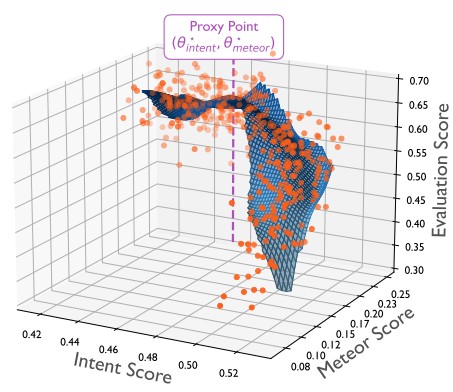

Figure 3.2: Correlation influences proxy points.

versity of text outputs and averaged them to serve as our evaluation metric (details in Appendix A). Our choice is in line with prior work that uses held out metrics as the ground truth for convenience of iteration (Gao et al., 2022). We call the value at which further increasing the proxy reward results in decreased ground-truth performance the *proxy point* $\theta^\star$. To identify proxy points, we trained PPO agents (Schulman et al., 2017) to maximize only one reward or the other (without KL regularization) and plotted the resulting evaluation scores against the METEOR and intent scores in Fig. 3.1. In both cases, the evaluation score initially increases before falling. Gao et al. (2022) also observed that, in general, maximization of reward causes the KL divergence between the trained and pretrained policies to increase, and therefore we also expect evaluation score to initially increase before decreasing as the KL grows as well, also shown in Fig. 3.1. One additional phenomenon that makes optimization of composite RMs challenging is that the component RMs may be correlated. We hypothesized that this interaction would influence the proxy points of the component rewards. To test this, we plotted the evaluation scores as a function of the METEOR and intent rewards for each run shown in Fig. 3.1 in Fig. 3.2 and fit a polynomial surface to the data, using kernel density estimation to only fit the surface over regions with sufficient data (further details in Appendix A). The maximizing point $(\theta_{intent}^\star, \theta_{meteor}^\star)$ indeed differs from the proxy points found by only considering one RM at a time. It is also important to note that the predicted maximizing point is of the fitted surface, rather than any point attained by one of the individual runs.

## 4 CONSTRAINED RLHF

Once one has identified proxy points for the component reward models, the next question is how to train agents to maximize these rewards until they hit their critical values. We propose that a useful approach to doing this is to reformulate the optimization objective using constraints.

**Adding Constraints to RL** In constrained reinforcement learning, an agent seeks to maximize its value while adhering to constraints on its behavior. Mathematically, this problem is formalized as a *constrained* MDP (CMDP; Altman, 1999), which is defined as a tuple $\mathcal{M}_C \triangleq \left(\mathcal{S}, \mathcal{A}, P, r_0, \gamma, \rho, \{r_i\}_{i=1}^N, \{\theta_i\}_{i=1}^N\right)$. Here, $\mathcal{S}, \mathcal{A}, P, r_0, \gamma,$ and $\rho$ are all as defined for standard MDPs (with $r_0$ the reward function), with $r_i : \mathcal{S} \times \mathcal{A} \to \mathbb{R},\ i = 1, \ldots, N$ being *constraint reward functions* and $\theta_i \in \mathbb{R},\ i = 1, \ldots, N$ associated *constraint thresholds*. Note that the subscripts on $r_{0:N}$ are indices over reward functions, not time steps. For clarity, we will hereafter refer to $r_0$ as the

"task reward" rather than just the reward. Rather than simply maximize value with respect to $r_0$, the CMDP optimization problem is given by

$$\max_{\pi} v_0^{\pi} \quad \text{s.t.} \quad v_i^{\pi} \geq \theta_i, \ i = 1, \dots, N. \tag{4.1}$$

That is, CMDPs represent behaviors which one would like to constrain in the form of value estimates with respect to reward functions which measure these behaviors. The $\geq$ symbol in Eq. (4.1) can easily be reversed if the constraint(s) encode behaviors which should be limited, and the inequality constraint(s) can be replaced with equality constraint(s). While there are many possible formulations, we default to the canonical form in Eq. (4.1) for the purposes of exposition.

**Proposed Method**   Given our possible objectives, we can now consider how to optimize them. One popular approach to solving constrained problems such as Eq. (4.1) is to use Lagrangian relaxation (Everett, 1963; Altman, 1999):

$$\max_{\pi} \min_{\boldsymbol{\mu} \geq 0} v_0^{\pi} + \sum_{i=1}^{N} \mu_i (v_i^{\pi} - \theta_i) \triangleq \mathcal{L}(\pi, \boldsymbol{\mu}), \tag{4.2}$$

where the weights on the value of each RM $\boldsymbol{\mu} = [\mu_1, \dots, \mu_N]^{\mathsf{T}} \in \mathbb{R}_{\geq 0}^{N}$ are the Lagrange multipliers associated with each constraint. In the case that we use equality constraints rather than inequality constraints, we use the variable $\boldsymbol{\xi}$ rather than $\boldsymbol{\mu}$. Optimization then proceeds by collecting experience using the policy and updating the policy and Langrange multipliers using gradient descent-ascent. We stress that the Lagrange multipliers are *not* fixed hyperparameters, but rather are learned as part of the optimization process. The negative gradient with respect to $\boldsymbol{\mu}$ is simply the constraint violation: $-\nabla_{\mu_i} \mathcal{L}(\pi, \boldsymbol{\mu}) = \theta_i - v_i^{\pi}$. To see how policy optimization works, we can rewrite the Lagrangian as

$$
\begin{aligned}
\mathcal{L}(\pi, \boldsymbol{\mu}) &= v_0^{\pi} + \sum_{i=1}^{N} \mu_i v_i^{\pi} - \sum_{i=1}^{N} \mu_i \theta_i \\
&= \mathbb{E}_{\substack{s_0 \sim \rho(\cdot) \\ a_0 \sim \pi(\cdot|s_0)}} \left[ q_0^{\pi}(s_0, a_0) + \sum_{i=1}^{N} \mu_i q_i^{\pi}(s_0, a_0) \right] - \sum_{i=1}^{N} \mu_i \theta_i \\
&= \mathbb{E}_{\substack{s_0 \sim \rho(\cdot) \\ a_0 \sim \pi(\cdot|s_0)}} \left[ q_{\boldsymbol{\mu}}^{\pi}(s_0, a_0) \right] - \sum_{i=1}^{N} \mu_i \theta_i,
\end{aligned}
\tag{4.3}
$$

where we define $q_{\boldsymbol{\mu}}^{\pi}(s, a) \triangleq q_0^{\pi}(s, a) + \sum_{i=1}^{N} \mu_i q_i^{\pi}(s, a)$ as the *mixed* $q$-values of policy $\pi$ given the current Lagrange multipliers $\boldsymbol{\mu}$. Note that this value is non-stationary, as the same policy will have a different value as the weightings on each constraint value change. Policy optimization then proceeds as normal with respect to the mixed $q$-values. As is frequently done in deep RL to reduce variance, we can replace the mixed $q$-values with mixed *advantages* $A_{\boldsymbol{\mu}}^{\pi} \triangleq q_{\boldsymbol{\mu}}^{\pi}(s, a) - v_{\boldsymbol{\mu}}(s)$, with $v_{\boldsymbol{\mu}}(s) = \mathbb{E}_{a \sim \pi} q_{\boldsymbol{\mu}}(s, a)$. We can optimize this objective with any policy gradient approach, in our case PPO. Detailed pseudocode is provided in Algorithm 1.

**Formal Guarantees**   While our focus is primarily empirical, we briefly comment on the theoretical properties of the above approach. Lagrangian relaxation converts the CMDP problem into a min-max game. If the values are decomposed as $v_i^{\pi} = \langle r_i, d_{\pi} \rangle$, where $d_{\pi}(s, a) \triangleq (1 - \gamma) \sum_{t \geq 0} \Pr(s_t = s, a_t = a | \pi)$ is the policy's cumulative, discounted state-action occupancy measure, and optimization is performed over $d_{\pi}$, then the problem is convex-concave and gradient descent-ascent (under basic assumptions) guarantees convergence of the average iterates to a saddle point, *i.e.*, $\left( K^{-1} \sum_{k=1}^{K} d_{\pi}^{(k)}, K^{-1} \sum_{k=1}^{K} \mu^{(k)} \right) \to (d_{\pi}^{\star}, \mu^{\star})$ as the number of iterations $K \to \infty$ (Freund and Schapire, 1997). However, in large-scale problems it is difficult to optimize directly over $d_{\pi}$, and we instead update the policy directly. In this case, the problem is convex in $\boldsymbol{\mu}$ but non-concave in $\pi$. Efroni et al. (2020) show sublinear regret bounds with respect to both policy optimality and constraint satisfaction using an optimistic approach, and Ding et al. (2020) show a convergence rate for the averaged iterates for general smooth policy classes of $\mathcal{O}(1/\sqrt{K})$ for the policy and $\mathcal{O}(1/K^{1/4})$ for the constraint violation using natural policy gradients. There is significant work on primal-dual policy optimization for CMDPs, which we discuss further in Appendix C.

| Method | Objective | Intuition |
|---|---|---|
| PPO (no KL) | $\max_\pi \sum_i \alpha_i v_i^\pi$ | Max. values |
| PPO | $\max_\pi \sum_i \alpha_i v_i^\pi$ s.t. $v_{\mathsf{KL}}^\pi \geq \theta_{\mathsf{KL}}$ | Max. values & stay close to $\pi_0$ |
| | New Methods | |
| PPO-SAT | Find $\pi \in \{\pi | v_i^\pi = \theta_i \, \forall i\}$ | Find 'feasible' $\pi$ s.t. values hit targets |
| $\mu$-PPO | $\max_\pi v_{\mathsf{KL}}^\pi$ s.t. $v_i \geq \theta_i \, \forall i$ | Stay close to $\pi_0$ s.t. RMs high enough |
| All-PPO | $\max_\pi \sum_i \alpha_i v_i^\pi$ s.t. $v_i \leq \theta_i \, \forall i$, $v_{\mathsf{KL}}^\pi \geq \theta_{\mathsf{KL}}$ | Max. RMs but not too much |
| $\xi$-PPO | $\max_\pi v_{\mathsf{KL}}^\pi$ s.t. $v_i = \theta_i \, \forall i$ | Stay close to $\pi_0$ & ensure RMs hit targets |

Table 1: A summary of the approaches we consider.

**Choosing a Constrained Objective**   Given this approach, we can now consider possible constraint formulations, all of which should embody the intuition that the agent should maximize each component reward only until its corresponding proxy point. This naturally suggests that the proxy points should be used as thresholds in the constrained objective. However, there are a number of possible formulations to consider when casting RLHF as a CMDP with this goal in mind. Once the proxy point for a given RM is reached, the agent has two options: continue to update the Lagrange multiplier on that RM to ensure that values remain at that point (via equality constraints), or simply stop optimizing/un-weight that RM entirely, *i.e.*, set the multiplier to zero, only re-weighting it if the constraint is violated (via inequality constraints). This latter approach carries that risk that the value with respect to that RM will continue to increase (past the proxy point) as other RMs continue to be optimized, but may be empirically effective if this is not the case and optimization is simplified by having a source of non-stationarity eliminated. In both of these cases, each component RM is assigned a constraint threshold, but the question of how to set the task reward remains. We propose the *KL reward* $r_{\mathsf{KL}} = -\log \frac{\pi(a_t|s_t)}{\pi_0(a_t|s_t)}$ as the main task reward. Gao et al. (2022) liken the KL to a resource which the agent spends, such that it should try to maximize its reward while limiting its divergence from the original policy as much as possible. Using the negative KL as the task reward carries the intuition of keeping the policy as similar as possible to the pretrained policy, subject to the constraint that each RM hits the point beyond which it stops aligning with the true objective. Note that the requirement that the agent hits these thresholds is crucial, as it prevents the agent from fully maximizing the negative KL reward (*i.e.*, remaining at the pretrained policy). In addition to these, there is another possible constrained approach wherein the agent simply maximizes the combined reward as in standard PPO (with KL regularization), but constrained so that each individual RM does not violate its respective threshold. All of these methods can be implemented by different settings of Algorithm 1. Finally, one could try to formulate the problem as one purely of constraint satisfaction: find any feasible policy whose values with respect to each of the RMs hit the appropriate proxy points. This could be implemented via a reward function that penalizes deviations from these points, *e.g.*, $r_{\mathsf{SAT}} = -\sum_i \alpha_i (r_i - \theta_i)^2$. However, this approach (Algorithm 2) faces the same problem as standard PPO—namely, how to best set the weights $\alpha_i$. These approaches are summarized in Table 1.

**Hacks**   Here, we describe several practical modifications to the "ideal" algorithm which we found to improve empirical performance. In practice, the noise and non-stationarity that primal-dual optimization in RL must contend with can lead to instability in the updates for the Lagrange multipliers. To handle this in practice, we follow prior work (Stooke et al., 2020; Zahavy et al., 2022; Moskovitz et al., 2023a) and use a sigmoid function to bound the Lagrange multipliers between 0 and 1. This results in mixed advantages which are a convex combination of the task and constraint advantages:

$$A_{\boldsymbol{\mu}}^\pi(s, a) = \left( N - \sum_{i=1}^{N} \sigma(\mu_i) \right) A_0^\pi(s, a) + \sum_{i=1}^{N} \sigma(\mu_i) A_i^\pi(s, a). \tag{4.4}$$

This equation has the intuitive interpretation of placing more weight on optimizing constraint reward $r_{i>0}$ when $\mu_{i>0}$ is high (indicating a constraint violation), and more weight on task reward $r_0$ when $\mu_{1:N}$ are low (indicating that constraints are satisfied). When we use equality constraints rather than inequality constraints, we replace the sigmoid with a $\tanh$ function (bounding the Lagrange multipliers between $-1$ and $1$). When updating the Lagrange multipliers, we found that using low or no momentum in the optimizer (we use SGD with a momentum parameter of 0.1) was helpful for

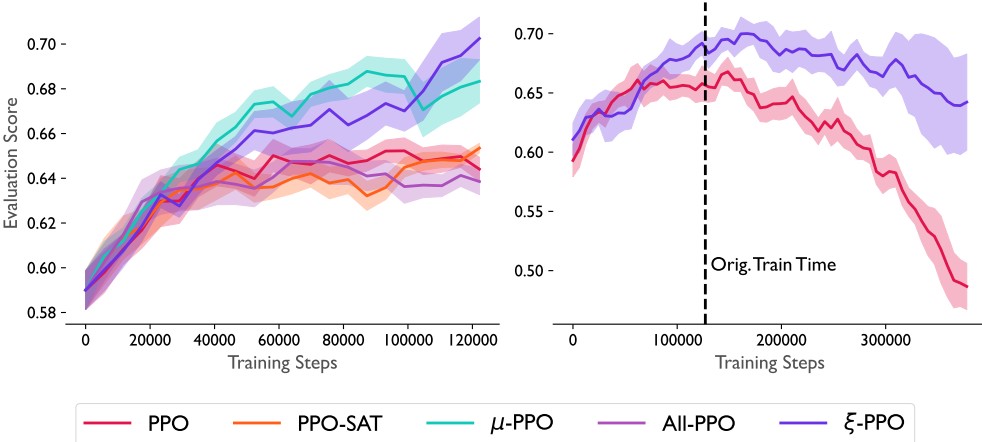

Figure 5.1: **Constrained RLHF improves evaluation performance.** (Left) Two constrained methods, $\mu$-PPO and $\xi$-PPO produce the best performance over the course of training. (Right) Balancing RMs using constraints makes performance more robust to longer training time.

performance, as otherwise $\sigma(\mu_i)$ or $\tanh(\xi_i)$ could be overly "sticky," remaining high for too long when constraints became satisfied and vice versa. Another hack which we found to be useful was to replace the value estimates in the constraint violation calculations with the sum of rewards to-go (for the appropriate reward function) for the remainder of a given episode. This is because we found that early in training, value estimates are inaccurate, which can cause the agent to incorrectly believe it is either adhering to or violating the constraint, leading to incorrect weighting of rewards via the Lagrange multiplier and slower overall learning.

## 5 EXPERIMENTAL EVALUATION

We now evaluate these possible approaches in the same setting as described in Section 3. The primary questions we would like to answer are as follows. (1) Do constrained methods result in better evaluation performance compared to PPO (and PPO-SAT)? (2) Do these approaches successfully enforce the desired constraints? (3) Do the thresholds determined by the proxy points lead to the best performance? Unless otherwise noted, all experiments are run for 5 random seeds, and any shading in plots denotes standard error. Code for all methods is available here: github.com/tedmoskovitz/ConstrainedRL4LMs.

**Does constrained RLHF improve performance?** In Fig. 5.1, we indeed find that two constrained approaches, $\mu$-PPO and $\xi$-PPO achieve better evaluation performance than other methods, with $\xi$-PPO performing slightly better at the end of training. To ensure fairness across methods, to set the fixed RM weightings used to train PPO and PPO-SAT, we selected the best settings found after 10 initial runs of each approach, the same as the total number of runs used to find proxy points used for the constrained methods. We conjecture that the strong performance of $\mu$- and $\xi$-PPO is due to the beneficial effects of jointly optimizing the policy and Lagrange multipliers (RM weightings). For example, even setting the weightings to be the *optimal* Lagrange multipliers and fixing them throughout training is not guaranteed to converge to a saddle point (Szepesvári, 2020), a phenomenon observed empirically by Moskovitz et al. (2023a). Notably, All-PPO did not perform as well as the other constrained methods, which we believe was due to increased instability in the optimization process (Appendix Fig. D.3). This is common in constrained problems with "paradoxical" objectives (Moskovitz et al., 2023a). Another benefit of continually modulating the weightings among RMs is that the weightings themselves are not hyper-optimized to a particular training duration. We trained both PPO and $\xi$-PPO using their hyperparameter settings optimized over runs with 128,000 steps for 3 times as long over 3 seeds and confirmed that the constrained approach was more stable (Fig. 5.1).

**Are constraints successfully enforced?** To verify that the constrained algorithms are working as expected, we plotted the intent and METEOR rewards across training for $\mu$-PPO, All-PPO, and $\xi$-PPO in Fig. 5.2. We can see that, as required by the constraints, $\mu$-PPO (approximately) reaches at least as high as the proxy point thresholds, All-PPO remains below them, and $\xi$-PPO approximately

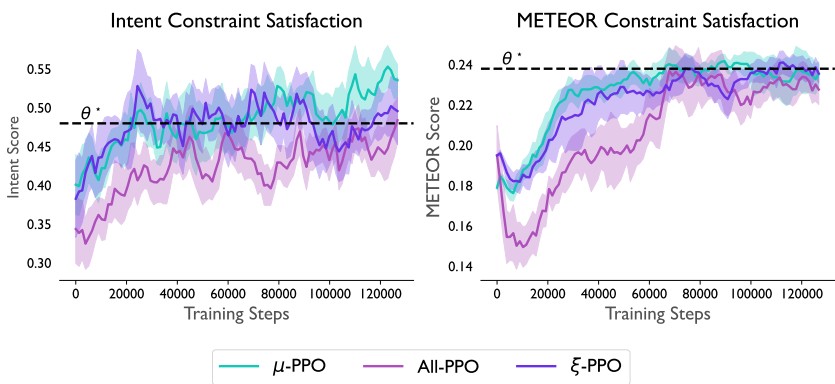

Figure 5.2: **Constraints are satisfied.** $\mu$-PPO reaches or exceeds the required intent (left) and METEOR (right) thresholds (dashed lines), All-PPO remains below them, and $\xi$-PPO hits them.

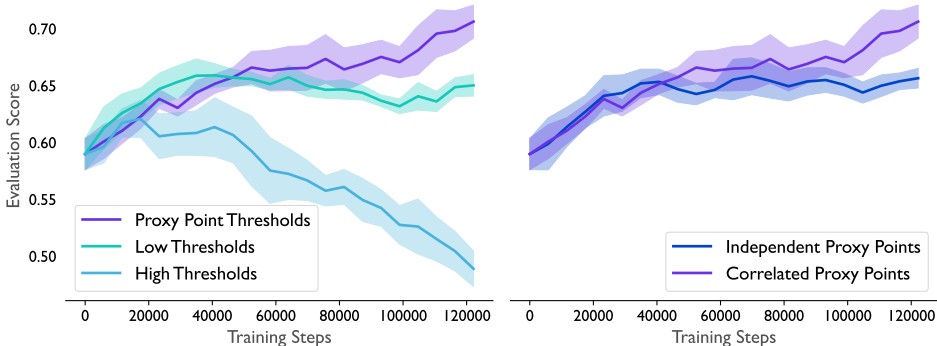

Figure 5.3: **Using proxy points as thresholds leads to the best performance.** (Left) Using thresholds that are 10% lower or higher reduces performance compared to proxy point thresholds. (Right) The proxy points that account for the correlation between RMs are more effective than those estimated independently.

hits them. $\mu$-PPO continues to increase above the intent proxy point, which may contribute to its slightly worse final performance compared to $\xi$-PPO in Fig. 5.1.

**Are proxy points the best thresholds?**   We compared the performance of $\xi$-PPO using the proxy points identified in Section 3 against the same method using thresholds that were 10% lower and 10% higher. The left panel of Fig. 5.3 shows that making thresholds lower causes initial performance to increase more quickly, as once the easier-to-reach thresholds are met, the agent is able to begin tightening the KL with respect to the pretrained policy earlier. However, performance plateaus at a lower level. When thresholds are set too high, the KL reward is ignored and the proxy rewards are optimized beyond the point at which they are useful, leading to worse performance. We also compared the performance of $\xi$-PPO using the correlated proxy points found in Fig. 3.2 against the independent proxy points found by only considering one RM at a time (Fig. 3.1).

## 5.1 IMPROVING THRESHOLD IDENTIFICATION

One downside of all methods considered so far is the need for multiple runs to either select a fixed weighting of RMs or identify proxy points. It would save significant compute—and reduce environmental impact, particularly for larger models—if it were possible to identify thresholds over the course of a single training run. Assuming we are allowed a limited number of queries to the evaluation metric over the course of training, one approach to accomplishing this would be to use a gradient-free optimizer to update the constraint thresholds to reach better performance. In order to limit the required number of policy updates between threshold updates, we used a local hill-climbing algorithm, Nelder-Mead (Nelder and Mead, 1965), which iteratively updates a simplex of thresholds based on the evaluation performance at each point. Once a new set of thresholds is proposed, we

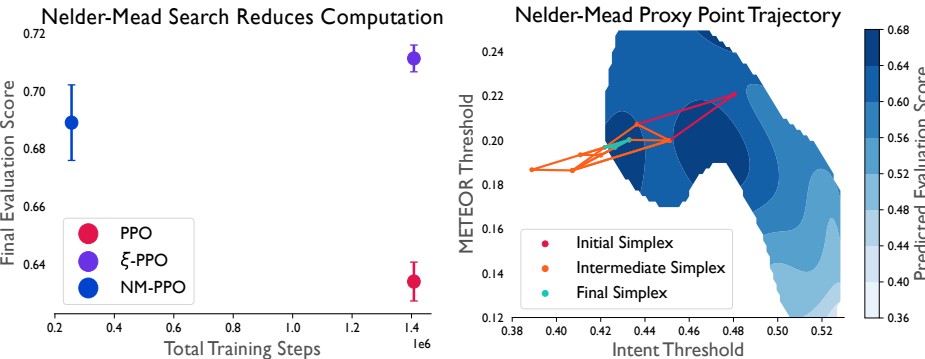

Figure 5.4: **Nelder-Mead threshold search saves computation.** (Left) Final evaluation performance versus total number of training steps (including hyperparameters searches). We allowed NM-PPO twice as many training steps for a single run, 256,000. (Right) An example threshold simplex trajectory overlaid on a contour plot of predicted evaluation performance from Fig. 3.2. The search converges to a local maximum.

use ξ-PPO to converge to those points and then evaluate the model once they're reached. Details are provided in Appendix A.4. We plotted the final evaluation performance of this variant of our approach, which we term NM-PPO (Algorithm 4), versus total number of training steps (including runs used for hyperparameter optimization) of PPO and ξ-PPO in Fig. 5.4. We found that NM-PPO obtains strong performance over the course of a single run, significantly saving in computation. Furthermore, the trajectories of simplexes proposed by Nelder-Mead closely follow the predicted evaluation performance found in Fig. 3.2, converging to local maxima of the surface. In Fig. 5.4, the trajectory converges to a local maximum rather than the global maximum, though other runs did indeed find the global optimum as predicted by Fig. 3.2 (Appendix Fig. D.5). One caveat with respect to this result is that the feasible region of threshold pairs is relatively small. There is therefore a moderate chance that the initial simplex already contains at least one threshold pair which produces reasonable performance. Further experimentation is required on problems with larger feasible regions and more than two component RMs.

# 6 DISCUSSION

In this work, we studied reward model overoptimization and the influence of correlation on proxy points in composite RMs. Then, we introduced a set of approaches for identifying and using these points as thresholds within a constrained optimization approach to RLHF. One weakness shared by all approaches—unconstrained and constrained alike—is that at least some minimal degree of access to the true objective/evaluation metric is required. Though in resource-rich settings this could be feasible (*e.g.*, by occasionally freezing training and querying human evaluators or using AI feedback), ideally, this would be dispensed with entirely. However, doing so is beyond the scope of this work. One weakness of gradient descent-ascent applied to primal-dual policy optimization is that it does not guarantee that the final policy and Lagrange multiplier(s) converge to a saddle point, only their averages. It would be an interesting direction for future work to apply an approach which does have such guarantees, such as ReLOAD (Moskovitz et al., 2023a). For optimizing the constraint thresholds during a single run, it would be interesting to explore alternative optimizers to Nelder-Mead, such as Bayesian optimization. Another interesting direction for future work would be to study the usefulness of a CMDP formulation for avoiding degeneration/collapse of model outputs, as while a deterministic optimal policy always exists for standard MDPs, CMDPs may demand optimal policies which are stochastic (Szepesvári, 2020). A similar idea was explored using a maximum entropy formulation by Khalifa et al. (2020). In general, further testing of our methods is necessary on more domains and with composite RMs with more components. We believe there are additional interesting avenues to explore in mitigating overoptimization, such as multi-objective RL (Abdolmaleki et al., 2020) or with constraints added to supervised learning (Rafailov et al., 2023). More broadly, we believe constrained optimization offers an important toolbox for approaching the alignment problem.

**Acknowledgements** Ted Moskovitz is funded by the Gatsby Charitable Foundation. Tuomas Sandholm is supported by the Vannevar Bush Faculty Fellowship ONR N00014-23-1-2876, National

Science Foundation grants RI-2312342 and RI-1901403, and ARO award W911NF2210266. Stephen McAleer is funded by a CI Fellowship. The authors would like to thank Vivek Veeriah, Tom Zahavy, Misha Laskin, and Dave Abel for helpful discussions.

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

## A    EXPERIMENTAL DETAILS

### A.1    SETTING

We use the same general experimental setting as Ramamurthy et al. (2022). The context window was of length 5, and separating the conversations in this way resulted in 35k training, 3k validation, and 3k test utterances. As in Ramamurthy et al. (2022), we use top-$k$, $k = 20$ sampling for decoding. The inputs to the model are concatenated snippets of human conversation in which changes of speaker are denoted by a special end-of-utterence (<EOU>) token. The intent classifier reward was derived from a finetuned RoBERTa model (Liu et al., 2019) which awards a score of 1 if the classified intent of the model's utterance matches that of the reference/ground truth utterance and 0 otherwise.

### A.2    THE EVALUATION METRIC

As we note in the main text, our objective in constructing an evaluation metric was to find one for which Goodhart's Law holds with respect to both the METEOR and intent reward functions, not to directly model human preferences. We therefore chose three metrics measuring lexical quality and three metrics measuring text diversity from among the metrics available in the RL4LMs codebase published by Ramamurthy et al. (2022). Specifically, the lexical metrics we used were SACREBLEU $x_s$ (Post, 2018), ROUGE2 $x_r$ (Lin, 2004; Ganesan, 2018), and BLEU $x_b$ (Papineni et al., 2002), and the diversity metrics we used were `unique-3` $x_u$, `vocab_size-3-nopunct` $x_v$, and `max_pred_length-nopunct` $x_m$. For each metric, we individually normalized the score between 0 and 1 (based on the range of observed values across all runs of PPO - METEOR and PPO - Intent), then averaged the resulting lexical scores and resulting diversity scores, before averaging the two average category scores. More precisely:

$$\texttt{eval\_score} = \frac{1}{2}\left( \frac{x_s + x_r + x_b}{3} + \frac{x_u + x_v + x_m}{3} \right). \tag{A.1}$$

Given the use of GPT2 on a relatively structured Dialog dataset, we opted for text overlap metrics for relevance, supplemented by diversity metrics Khalifa et al. (2020). Future work on larger models and more involved tasks (e.g., summarization Stiennon et al. (2020)) could consider more sophisticated evaluation methodology, taking inspiration from works such as Maynez et al. (2023). For this work, we focused on a simple evaluation metric designed to facilitate our study of overoptimization, rather than to emulate human ratings or rank different PLMs.

### A.3    FITTING THE EVALUATION SCORE SURFACE

The overall procedure is described in Phase 1 of Algorithm 3, where $\mathcal{F}$ is the function class for the evaluation score estimator. In our case, $\mathcal{F}$ was the space of polynomials of degree 10. To avoid predicting high evaluation scores over regions of the METEOR × intent space with little or no data points, we employed kernel density estimation with a Gaussian kernel to create a mask which hid parts of the fitted surface over low-density data regions (with a threshold density of 50/square unit). This approach is purely heuristic and could likely be greatly improved on in future work. To find the argmax of the surface, we tried two approaches. First, we used the built-in implementation of L-BFGS-B Zhu et al. (1997) in the SciPy `optimize` package (Virtanen et al., 2020) using the results from KDE to provide bounds. Second, since the polynomial surface is approximated using a discrete mesh, we also simply picked the coordinates corresponding to the plotted maximum. There was a negligible difference between methods, so we used the second approach in practice.

### A.4    NELDER-MEAD PPO DETAILS

We provide detailed pseudocode of our approach in Algorithm 4. In practice, we found several implementation details to be important for ensuring good performance. First, the initial simplex was crucial. Rather than initialize thresholds randomly across the entire range of possible METEOR and intent values, we initialize them based on random perturbations of the evaluation of the initial/pretrained policy (i.e., what the METEOR and intent scores are at the beginning of finetuning). This was very helpful, as otherwise Nelder-Mead would propose threshold pairs that were effectively not feasible for the policy to achieve, e.g., a very high METEOR threshold with a very low intent threshold. Second, we capped the number of iterations allowed for one evaluation/threshold setting at 1/8 of the total allowed training steps. Without this, the agent would often waste most of its run trying to hit challenging/infeasible thresholds. If the thresholds couldn't be reached in that time, the evaluation score was computed wherever the agent was at that time. Third, the agent cached the eval scores of previously-reached threshold pairs—if Nelder-Mead proposed a threshold pair that had

been reached before (or is within a elementwise tolerance of $\pm 5\%$ of a previously-reached pair) then it just returns the evaluation score it measured previously rather than updating the policy to return to it. The Nelder-Mead hyperparameters we use are $\alpha = 1, \gamma = 2, \rho = 0.5, \sigma = 0.5$—these settings are untuned, and could likely be adjusted to improve performance.

### A.5 COMPUTATIONAL RESOURCES

All experiments were performed on a single NVIDIA A100 GPU, with each run taking between 8 and 10 hours with the exception of runs for Nelder-Mead PPO, which took approximately 20 hours.

### A.6 ALGORITHM HYPERPARAMETERS

| Hyperparameter | PPO | PPO-SAT | $\mu$-PPO | All-PPO | $\xi$-PPO |
|---|---|---|---|---|---|
| Steps per Update ($M'$) | 1,280 | 1,280 | 1,280 | 1,280 | 1,280 |
| Total Steps ($KM'$) | 128,000 | 128,000 | 128,000 | 128,000 | 128,000 |
| Batch Size ($B$) | 64 | 64 | 64 | 64 | 64 |
| Epochs per Update ($L$) | 5 | 5 | 5 | 5 | 5 |
| Learning Rate ($\eta$) | 1e-6 | 1e-6 | 1e-6 | 1e-6 | 1e-6 |
| Initial KL Coefficient ($\alpha_0$) | 0.2 | 0.2 | 0.2 | 0.2 | 0.2 |
| Target KL | 0.5 | 0.5 | - | 0.5 | - |
| Discount Factor ($\gamma$) | 0.99 | 0.99 | 0.99 | 0.99 | 0.99 |
| GAE $\lambda$ | 0.95 | 0.95 | 0.95 | 0.95 | 0.95 |
| Clip Ratio ($\epsilon$) | 0.2 | 0.2 | 0.2 | 0.2 | 0.2 |
| Rollouts Top-$k$ | 20 | 20 | 20 | 20 | 20 |
| Value Function Coefficient ($\alpha_v$) | 0.5 | 0.5 | - | - | - |
| METEOR Coefficient ($\alpha^{met}$) | 0.5 | 0.5 | - | - | - |
| Intent Coefficient ($\alpha^{int}$) | 1.0 | 1.0 | - | - | - |
| METEOR Proxy Point ($\theta^\star_{meteor}$) | - | - | 0.23 | 0.23 | 0.23 |
| Intent Proxy Point ($\theta^\star_{intent}$) | - | - | 0.48 | 0.48 | 0.48 |
| METEOR Value Coefficient | - | - | 0.5 | 0.5 | 0.5 |
| Intent Value Coefficient | - | - | 0.5 | 0.5 | 0.5 |
| KL Value Coefficient | - | - | 0.2 | - | 0.2 |
| Lagrange Multiplier Function | - | - | sigmoid | sigmoid | tanh |

Table 2: Experiment Hyperparameters.

## B THE KL REGULARIZATION COEFFICIENT

As introduced by Ziegler et al. (2019), it is common in RLHF with PPO to adapt the KL coefficient $\alpha^{\mathsf{KL}}$ with the following update:

$$e_t = \mathrm{clip}\left(\frac{\mathsf{KL}[\pi(\cdot|s_t); \pi_0(\cdot|s_t)] - \theta^{\mathsf{KL}}}{\theta^{\mathsf{KL}}}, -0.2, 0.2\right)$$

$$\alpha^{\mathsf{KL}}_{t+1} = \alpha^{\mathsf{KL}}_t(1 + \eta^{\mathsf{KL}} e_t),$$

where $\theta^{\mathsf{KL}}$ is a hyperparameter which effectively acts as an upper limit on the KL from the initial policy, and $\eta^{\mathsf{KL}}$ acts like a learning rate. The KL coefficient then follows the path of a Lagrange multiplier with $\theta^{\mathsf{KL}}$ as its constraint threshold, as the constraint violation $\mathsf{KL}[\pi(\cdot|s_t); \pi_0(\cdot|s_t)] - \theta^{\mathsf{KL}}$ is exactly the gradient with respect to such a Lagrange multiplier.

## C ADDITIONAL RELATED WORK

In addition to the discussion in the main text, there is a long history of work on CMDPs. Borkar (2005) first studied actor-critic approaches in this context, and Bhatnagar and Lakshmanan (2012) were the first to consider constrained policy optimization with function approximation. More broadly, Achiam et al. (2017), Chow et al. (2018), Paternain et al. (2019), Tessler et al. (2019), Calian et al. (2020), Efroni et al. (2020), Stooke et al. (2020), Moskovitz et al. (2023a), and Ding and Lavaei (2023) all study the problem of integrating constraints into RL. More generally, an important factor

---

Algorithm 1: Constrained PPO for Dialogue Generation

---

1: **Require**: Dataset $\mathcal{D} = \{(\mathbf{x}^m, \mathbf{y}^m)\}_{m=1}^M$, initial policy parameters $\psi^{(1)}$, initial parameters for value functions $\phi_0^{(1)}, \ldots, \phi_N^{(1)}$, constraint thresholds $\theta_1^{(1)}, \ldots, \theta_N^{(1)}$, initial Lagrange multipliers $\boldsymbol{\mu}^{(1)}$, squashing function $f \in \{\sigma, \tanh\}$

2: **for** step $k = 1, \ldots, K$ **do**

3:     // Sample experience

4:     Uniformly sample $M' < M$ contexts $\mathbf{x}^{m'} \sim \mathcal{U}(\mathcal{D})$

5:     Generate predicted 'trajectory' responses

$$\hat{\mathbf{y}}^{m'} = (a_1, \ldots, a_T) \sim p_\pi(\mathbf{y}) = \prod_{t=1}^T \pi(a_t|s_t)$$

    where $s_1 = \mathbf{x}^{m'}$

6:     Compute generalized advantage estimates:

$$(\hat{A}_i)_t^{(k)} = (\delta_i)_t + \gamma\mu(\delta_i)_{t+1} + \cdots + (\gamma\mu)^{T-t+1}(\delta_i)_{T-1}, \quad i = 0, \ldots, N,$$

    where $(\delta_i)_t \triangleq r_i(s_t, a_t, s_{t+1}, \mathbf{y}^{m'}) + \gamma\bar{v}_i(s_{t+1}) - v_i(s_t)$.

7:     Store advantages and trajectories in buffer $\mathcal{B}$

8:     // Update

9:     **for** epoch $\ell = 1, \ldots, L$ **do**

10:         **for** trajectory batch $\{((\hat{A}_{0:N})_b, (\delta_0)_b, \ldots, (\delta_N)_b, \hat{\mathbf{y}}_b)\}_{b=1}^B \sim \mathcal{U}(\mathcal{B})$ in $\mathcal{B}$ **do**

11:         Compute mixed advantage estimates:

$$(\hat{A}_{\boldsymbol{\mu}})_{bt}^{(k)} = \left(N - \sum_{i=1}^N f\left(\mu_i^{(k)}\right)\right)(\hat{A}_0)_{bt}^{(k)} + \sum_{i=1}^N f\left(\mu_i^{(k)}\right)(\hat{A}_i)_{bt}^{(k)}$$

12:         Compute the policy loss:

$$\mathcal{L}_{\text{PPO}} = -\frac{1}{BT}\sum_{b=1}^B\sum_{t=0}^{T-1}\min\{\rho_{bt}(\psi^{(k)})(\hat{A}_{\boldsymbol{\mu}})_{bt}^{(k)}, \text{clip}(\rho_{bt}(\psi^{(k)}), 1-\epsilon, 1+\epsilon)(\hat{A}_{\boldsymbol{\mu}})_{bt}^{(k)}\},$$

        where $\rho_{bt}(\psi^{(k)}) = \frac{\pi_{\psi^{(k)}}(a_{bt}|s_{bt})}{\pi_{\psi^{(k-1)}}(a_{bt}|s_{bt})}$.

13:         Compute the value function losses:

$$\mathcal{L}_{v_i} = \frac{1}{BT}\sum_{b=1}^B\sum_{t=0}^{T-1}\frac{1}{2}(\delta_i)_{bt}^2, \quad i = 0, 1 \ldots, N$$

14:         Update the policy and value functions via SGD on $\mathcal{L}_{\text{PPO}} + \alpha_v \sum_{i=0}^N \mathcal{L}_{v_i}$

15:         Update the Lagrange multipliers via SGD on $\mathcal{L}_\mu$:

$$\mathcal{L}_{\mu_i} = \frac{1}{BT}\sum_{b=1}^B\sum_{t=0}^{T-1}(v_i(s_{bt}) - \theta_i)f\left(\mu_i^{(k)}\right)$$

16:         **end for**

17:     **end for**

18:     Reset buffer $\mathcal{B} \leftarrow \varnothing$

19: **end for**

---

in using a Lagrangian approach to solving CMDPs is the introduction of non-stationarity into the reward function. RL with non-stationary rewards is an active area of interest in RL (Padakandla et al., 2020; Cheung et al., 2020; Lecarpentier and Rachelson, 2019), particularly in the context of continual RL (Khetarpal et al., 2022) often with some form of temporal structure introduced in the non-stationarity (Xie et al., 2020; 2021). An interesting case in additional to primal-dual optimization

---

Algorithm 2: PPO-SAT for Dialogue Generation

---

1: **Require**: Dataset $\mathcal{D} = \{(\mathbf{x}^m, \mathbf{y}^m)\}_{m=1}^M$, initial policy parameters $\psi^{(1)}$, initial parameters for value function $\phi^{(1)}$, proxy point estimates $\theta_1, \ldots, \theta_N$, RM coefficients $\alpha_1, \ldots, \alpha_N$

2: **for** step $k = 1, \ldots, K$ **do**

3:      // Sample experience

4:      Uniformly sample $M' < M$ contexts $\mathbf{x}^{m'} \sim \mathcal{U}(\mathcal{D})$

5:      Generate predicted 'trajectory' responses

$$\hat{\mathbf{y}}^{m'} = (a_1, \ldots, a_T) \sim p_\pi(\mathbf{y}) = \prod_{t=1}^T \pi(a_t|s_t)$$

     where $s_1 = \mathbf{x}^{m'}$

6:      Compute generalized advantage estimates:

$$\hat{A}_t^{(k)} = \delta_t + \gamma \delta_{t+1} + \cdots + (\gamma)^{T-t+1} \delta_{T-1}$$

     where $\delta_t \triangleq -\sum_{i=1}^N \alpha_i (r_i(s_t, a_t, s_{t+1}, \mathbf{y}^{m'}) - \theta_i)^2 + \gamma \bar{v}_{\phi^{(k)}}(s_{t+1}) - v_{\phi^{(k)}}(s_t)$.

7:      Store advantages and trajectories in buffer $\mathcal{B}$

8:      // Update

9:      **for** epoch $\ell = 1, \ldots, L$ **do**

10:         **for** trajectory batch $\{((\hat{A}_b, (\delta_0)_b, \ldots, (\delta_N)_b, \hat{\mathbf{y}}_b)\}_{b=1}^B \sim \mathcal{U}(\mathcal{B})$ in $\mathcal{B}$ **do**

11:            Compute the policy loss:

$$\mathcal{L}_{\text{PPO}}(\psi^{(k)}) = -\frac{1}{BT} \sum_{b=1}^B \sum_{t=0}^{T-1} \min\{\rho_{bt}(\psi^{(k)})\hat{A}_{bt}^{(k)}, \text{clip}(\rho_{bt}(\psi^{(k)}), 1-\epsilon, 1+\epsilon)\hat{A}_{bt}^{(k)}\},$$

           where $\rho_{bt}(\psi^{(k)}) = \frac{\pi_{\psi^{(k)}}(a_{bt}|s_{bt})}{\pi_{\psi^{(k-1)}}(a_{bt}|s_{bt})}$.

12:            Compute the value function loss:

$$\mathcal{L}_v(\phi^{(k)}) = \frac{1}{BT} \sum_{b=1}^B \sum_{t=0}^{T-1} \frac{1}{2} \delta_{bt}^2$$

13:            Update the policy and value functions via SGD on $\mathcal{L}_{\text{PPO}} + \alpha_v \mathcal{L}_v$

14:         **end for**

15:      **end for**

16:      Reset buffer $\mathcal{B} \leftarrow \varnothing$

17: **end for**

---

in which non-stationarity is introduced by the agent itself is in the use of epistemic uncertainty for more efficient exploration, manifested in the form of non-stationary exploration bonuses to reward (O'Donoghue, 2023; Tarbouriech et al., 2023). Non-stationarity may also be introduced as a means of modeling more naturalistic reward structures for studying animal behavior (Moskovitz et al., 2021a; 2023b). Finally, another area of related work is regularized policy optimization, whereby the standard reward-maximizing policy optimization objective is augmented with a regularization term, typically a divergence measure with respect to some reference policy (Berner et al., 2019; Espeholt et al., 2018). In the single-task setting, the updated policy is typically regularized to stay close to its current setting, which has close connections to natural gradient (Kakade and Langford, 2002; Moskovitz et al., 2021b; Pacchiano et al., 2020), trust region (Schulman et al., 2015), and variational inference (Levine, 2018; Haarnoja et al., 2018; Abdolmaleki et al., 2018) approaches. In the multitask setting, the policy is typically regularized towards some default policy which encodes behavior thought to be useful across a family of tasks, and which may be far from the current policy (Galashov et al., 2019; Teh et al., 2017; Moskovitz et al., 2022). This setting is quite similar in this sense to KL regularization as used in RLHF.

---

Algorithm 3: Two-Phase Approach

---

1: **Require:** Proxy RMs $r_{1:N} = r_1, \ldots, r_N$, Evaluation RM $r^\star$, policy gradient algorithm Alg, constrained algorithm CAlg (*e.g.*, Algorithm 1)
2: Phase 1: Proxy point identification
3: Evaluation RM dataset $\mathcal{D} \leftarrow \varnothing$
4: **for** $i = 1, \ldots, N$ **do**
5:     Fit Alg on $r_i$, collect $K$ measurements $\{(r_1, \ldots, r_N, r^\star)_k\}_{k=1}^K$ across training
6:     $\mathcal{D} \leftarrow \mathcal{D} \cup \{(r_1, \ldots, r_N, r^\star)_k\}_{k=1}^K$
7: **end for**
8: Fit evaluation RM predictor $f_r^\star$

$$f_r^\star \leftarrow \operatorname*{argmin}_{f_r \in \mathcal{F}} \frac{1}{NK} \sum_{i=1}^N \sum_{k=1}^K (r_{ik}^\star - \tilde{f}_r((r_1)_{ik}, \ldots, (r_N)_{ik}))^2$$

9: Proxy point: $\boldsymbol{\theta}^\star \leftarrow \operatorname{argmax}_{r_1, \ldots, r_N} f_r^\star(r_1, \ldots, r_N)$
10: Phase 2: Constrained optimization
11: $\pi^\star \leftarrow \mathsf{CAlg}(\boldsymbol{\theta}^\star)$
12: **Return** $\pi^\star$

---

Algorithm 4: Nelder-Mead Proxy Point Search

---

1: **Require:** Evaluation RM $r_{eval}$, initial simplex thresholds $\{\boldsymbol{\theta}_j \triangleq (\theta_{1:N})_j\}_{j=1}^{N+1}$, reflection coefficient $\alpha$, expansion coefficient $\gamma$, contraction coefficient $\rho$, shrinkage coefficient $\sigma$
2: Fit $\xi$-PPO using initial thresholds, compute $\{v_{eval}^\pi(\boldsymbol{\theta}_j)\}$
3: **while** not converged **do**
4:     Sort threshold sets by evaluation score $(\boldsymbol{\theta}_1, \ldots, \boldsymbol{\theta}_{N+1})$
5:     Compute the centroid of the $N$-best thresholds $\bar{\boldsymbol{\theta}} = \frac{1}{N} \sum_{i=1}^N \boldsymbol{\theta}_i$
6:     Reflect the worst point $\boldsymbol{\theta}_r = \bar{\boldsymbol{\theta}} + \alpha(\bar{\boldsymbol{\theta}} - \boldsymbol{\theta}_{N+1})$
7:     Fit $\xi$-PPO on the reflected thresholds and compute $v_{eval}^\pi(\boldsymbol{\theta}_r)$
8:     **if** $v_{eval}^\pi(\boldsymbol{\theta}_1) \leq v_{eval}^\pi(\boldsymbol{\theta}_r) < v_{eval}^\pi(\boldsymbol{\theta}_N)$ **then**
9:         $\boldsymbol{\theta}_{N+1} = \boldsymbol{\theta}_r$
10:         **GOTO** Line 3
11:     **end if**
12:     **if** $v_{eval}^\pi(\boldsymbol{\theta}_r) < v_{eval}^\pi(\boldsymbol{\theta}_0)$ **then**
13:         Expand: $\boldsymbol{\theta}_e = \bar{\boldsymbol{\theta}} + \gamma(\boldsymbol{\theta}_r - \bar{\boldsymbol{\theta}})$
14:         Fit $\xi$-PPO on the expanded thresholds and compute $v_{eval}^\pi(\boldsymbol{\theta}_e)$
15:         **if** $v_{eval}^\pi(\boldsymbol{\theta}_e) < v_{eval}^\pi(\boldsymbol{\theta}_1)$ **then**
16:             $\boldsymbol{\theta}_{N+1} = \boldsymbol{\theta}_e$
17:             **GOTO** Line 3
18:         **else**
19:             $\boldsymbol{\theta}_{N+1} = \boldsymbol{\theta}_r$
20:             **GOTO** Line 3
21:         **end if**
22:     **end if**
23:     Contract: $\boldsymbol{\theta}_c = \bar{\boldsymbol{\theta}} + \rho(\boldsymbol{\theta}_{N+1} - \bar{\boldsymbol{\theta}})$
24:     Fit $\xi$-PPO on the contracted thresholds and compute $v_{eval}^\pi(\boldsymbol{\theta}_c)$
25:     **if** $v_{eval}^\pi(\boldsymbol{\theta}_c) < v_{eval}^\pi(\boldsymbol{\theta}_{N+1})$ **then**
26:         $\boldsymbol{\theta}_{N+1} = \boldsymbol{\theta}_c$
27:         **GOTO** Line 3
28:     **end if**
29:     Shrink: $\boldsymbol{\theta}_j \leftarrow \boldsymbol{\theta}_1 + \sigma(\boldsymbol{\theta}_j - \boldsymbol{\theta}_1), \quad j = 2, \ldots, N+1$
30: **end while**

---

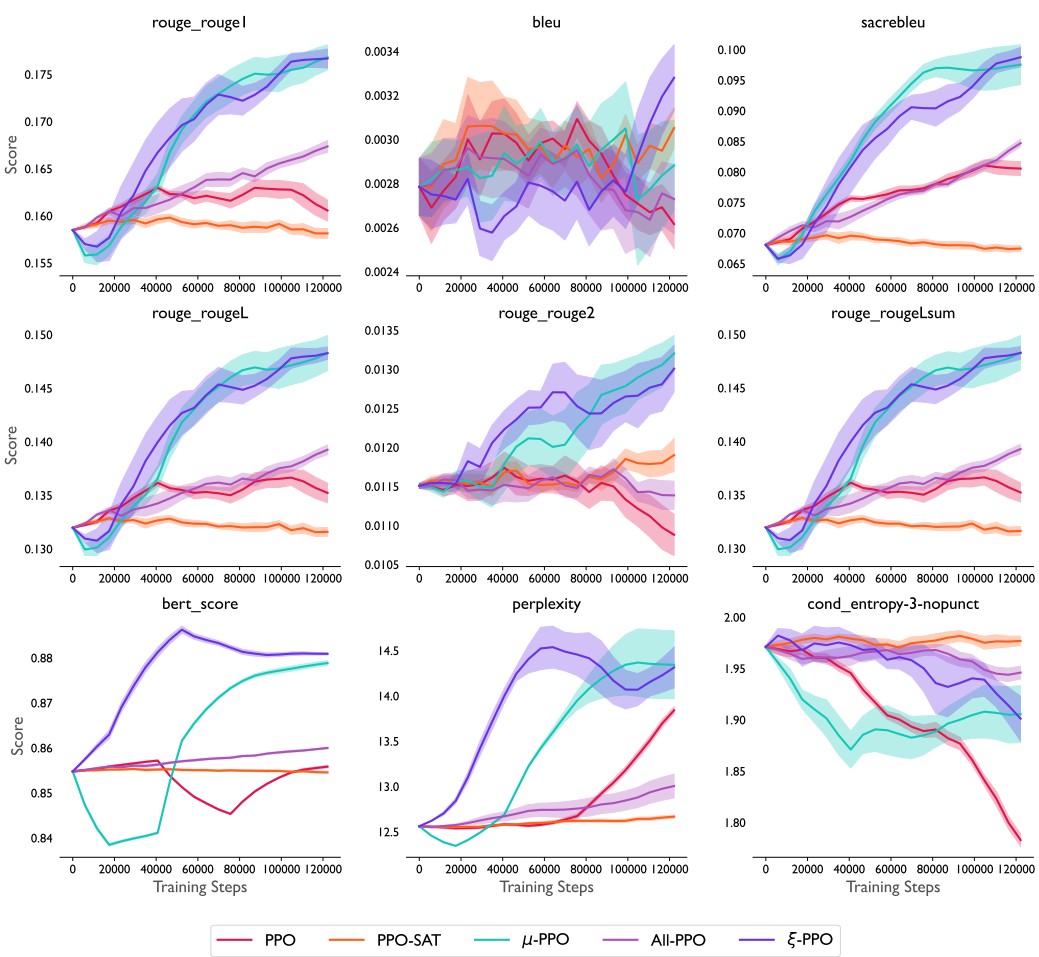

Figure D.1: **Performance of the tested methods across various metrics.**

# D    ADDITIONAL RESULTS

## D.1    ADDITIONAL METRICS

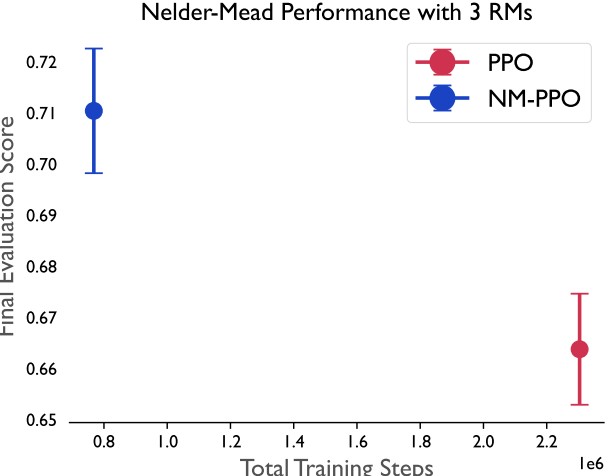

Figure D.2: **Nelder-Mead retains effectiveness with three component RMs.** To evaluate whether NM-PPO is able to scale effectively to more than two component RMs, we added BLEU score as a third RM and compared the performance to PPO. Because of the additional RM, we allowed PPO five additional tuning runs (15 total) to adjust the coefficients on the component RMs. Both NM-PPO and PPO (with the tuned coefficients) were then run for 3 seeds, with the means and standard errors plotted above. We can see the NM-PPO is again able to outperform PPO while using 1/3 the number of total training steps.

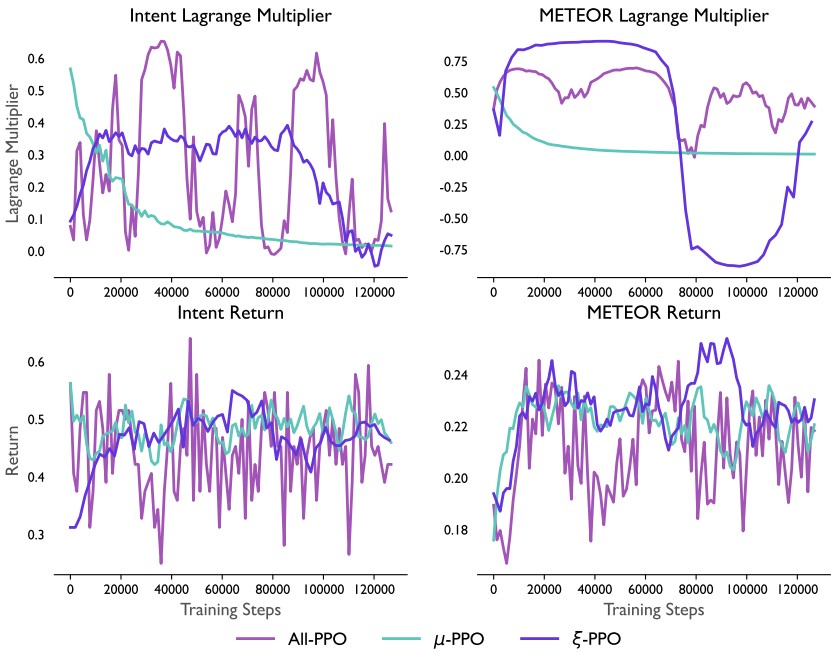

Figure D.3: **Paradoxical objectives and intermediate thresholds can worsen oscillations.** Running gradient descent-ascent on a min-max game only guarantees that the average of the iterates converges to the saddle point. In practice, this can mean that the Lagrange multiplier(s) and value(s) can oscillate wildly over the course of training, even if their averages converge. The problem is worse for constraint thresholds which are intermediate—those that are neither high nor low relative to the range of an individual reward function (Moskovitz et al., 2023a), but can be hidden by averaging. Above are example runs of All-PPO, $\mu$-PPO, and $\xi$-PPO, showing this problem is worse for All-PPO, likely due to its paradoxical objective (maximizing an objective which its constraints demand it minimize). This instability likely leads to worse performance. We can also see that $\xi$-PPO's METEOR Lagrange multiplier oscillates as well, though with long periods of stability, which likely mitigates the effect on performance.

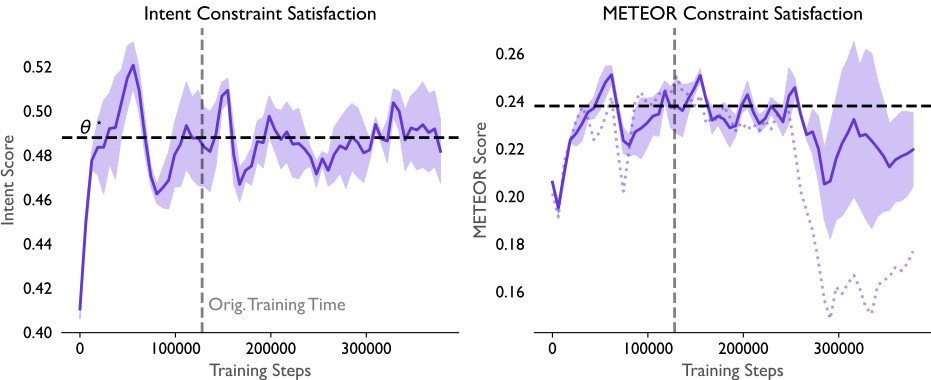

Figure D.4: $\xi$-**PPO constraint satisfaction for long training runs.** $\xi$-PPO is largely able to satisfy constraints with respect to its component RMs over the course of many training steps. However, it is possible for it to lose stability. As in other plots, solid curves and shading denote the means and standard errors across runs. In the right hand panel, the dotted curve indicates one training run which became unstable after roughly twice the length of the standard training time, with its METEOR score falling below the constraint threshold. This corresponded with a drop in evaluation performance, which reduced the overall average in Fig. 5.1.

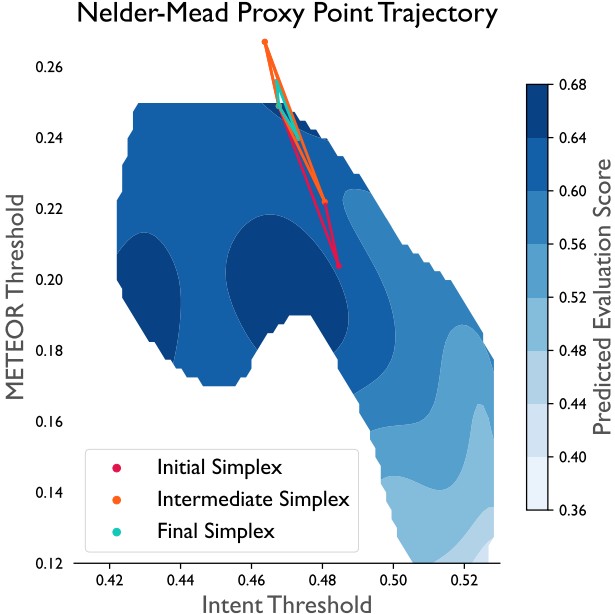

Figure D.5: **Another example Nelder-Mead simplex trajectory.** In this case, Nelder-Mead converged to the global maximum proxy point setting.

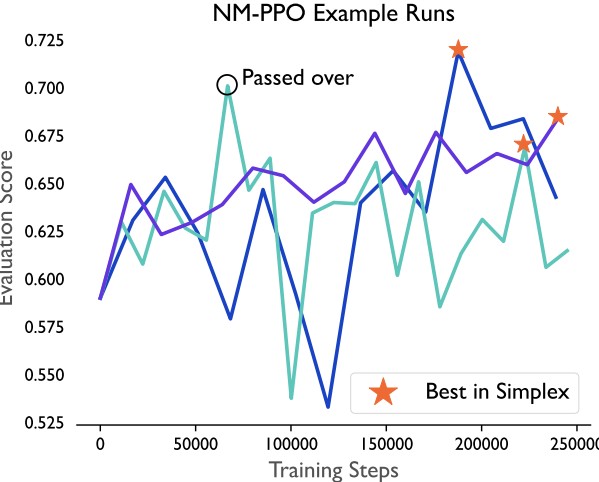

Figure D.6: **Several example NM-PPO evaluation curves.** This plot shows the evaluation scores across training of three runs of NM-PPO. There are several salient features to note. First, because of the nature of Nelder-Mead optimization, performance is not expected to be monotonic. Rather, the algorithm noisily hill-climbs as it adjusts the estimated proxy points comprising its current simplex. The algorithm returns the best simplex point, which may not be the final one tested. Second, not every evaluation point in the curves above corresponds to a query by the algorithm (i.e., upon reaching its target proxy point values), but is rather just measured for plotting purposes. This lets us see that occasionally, on the way to the next set of proxy points to query, the optimization process may induce the policy to "pass over" a region of high performance without being aware of it. This is what occurred at the circled point in the plot above.

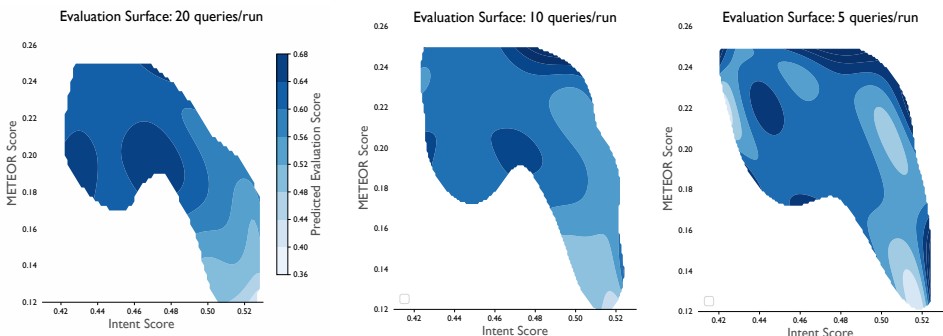

Figure D.7: **Predicted evaluation surface as a function of evaluation queries per run.** To visualize the effect of the number of queries to the evaluation function per run on the estimate of the evaluation function surface, we created contour plots of the predicted surface using the procedure described in Appendix A.3 for three different query frequencies: 20 (the default), 10, and 5. This was done using the same data used to generate Fig. 3.1. We can see that the basic structure of the predicted surface is preserved even down to 5 queries/run, and in particular that the global maximum is consistently located in the upper right region of the plots.

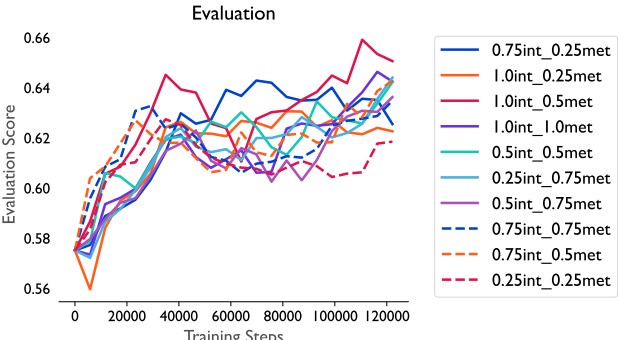

Figure D.8: **Sweep over possible RM weightings for PPO.**

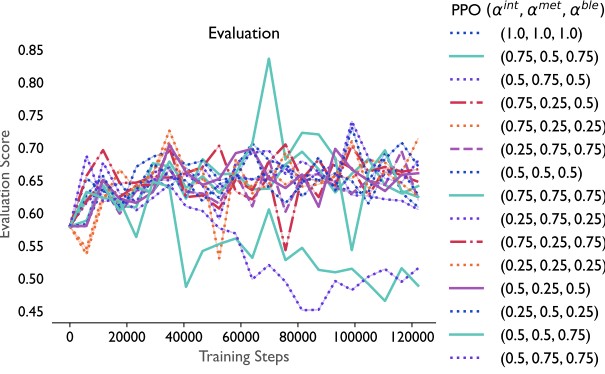

Figure D.9: **Sweep over possible RM weightings for PPO with 3 RMs (Intent, METEOR, and BLEU).**

### D.2  SAMPLE OUTPUTS

For reference, we provide several random prompts and model outputs for the various methods. The key takeaway is that high METEOR and intent scores do not necessarily correlate with high evaluation scores and vice versa.

**Prompt:**  `They didn ' t accept me .  <EOU> Are you serious ? <EOU> It ' s the truth .  <EOU> Are you okay ?  <EOU> I really wanted to get into NYU . <EOU>`

- **Reference:** `I am really sorry to hear that .  <EOU>`
- **PPO:**    `What were you trying to hide.  <EOU> I'm not hiding anything.  <EOU>`   METEOR:0.23, Intent: 1, Eval: 0.62
- **PPO-SAT:**    `What do you mean?  Did something happen to you? <EOU>`   METEOR:0.19, Intent: 1, Eval: 0.62
- **$\mu$-PPO:**    `What do you intend on doing in college?  <EOU> I don't know if I'll`   METEOR:0.22, Intent: 0, Eval: 0.66
- **All-PPO:** `What do you mean u don't accept people.  <EOU>`   METEOR:0.23, Intent: 1, Eval: 0.61
- **$\xi$-PPO:** `What did you do?  <EOU>`   METEOR:0.23, Intent: 0, Eval: 0.65
- **NM-PPO:** `If it wasn't a big enough place to be a professor, I could have stayed there.  <EOU>`   METEOR:0.22, Intent: 1, Eval: 0.66

**Prompt:**   `I can't .  I'm waiting for Paul , but he's late .  <EOU> I'll say the movie's starting in the minute .  <EOU> I know .  I wonder what's keeping him .  <EOU> Looks like he won't show up .  He's done this before , hasn't he ?  <EOU> Yeah , a number of times .  <EOU>`

- **Reference:** `What's he trying to pull anyway .  <EOU>`
- **PPO:**    `You're right.  I think he's been looking for Paul. <EOU> He said.`   METEOR:0.22, Intent: 1, Eval: 0.64
- **PPO-SAT:** `So we're waiting.  If you want to stay a bit longer you can go with Paul.  <EOU>`   METEOR:0.20, Intent: 1, Eval: 0.62
- **$\mu$-PPO:** `You're looking forward to meeting him tomorrow.  <EOU> I'll find out.  <EOU>`   METEOR:0.22, Intent: 1, Eval: 0.66
- **All-PPO:** `So we're waiting..  <EOU> And he just didn't show up. <EOU>`   METEOR:0.24, Intent: 1, Eval: 0.64
- **$\xi$-PPO:** `So I'm waiting.  <EOU> He's been doing this for a long time.`   METEOR:0.22, Intent: 1, Eval: 0.69
- **NM-PPO:**   `Oh no.  The second time, it was.  I can't even get into that stuff.  <EOU>`   METEOR:0.21, Intent: 1, Eval: 0.65

**Prompt:** `What's wrong with you ?  <EOU> I have a sore throat and headache .  <EOU> Do you feel tired ?  <EOU> Yes , my whole body feels weak .  And I really feel terrible .  <EOU> I'm sorry to hear that .  I'll have to examine you .  <EOU>`

- **Reference:** `Ok , go ahead .  <EOU>`
- **PPO:** `What's the matter?  You're always so busy.  <EOU>`   METEOR:0.20, Intent: 0, Eval: 0.64
- **PPO-SAT:** `What is your name?  <EOU> Moth <EOU> Moth <EOU>`   METEOR:0.17, Intent: 0, Eval: 0.60
- **$\mu$-PPO:**   `What's wrong with your food?  <EOU> No matter what.. I'll have to check`   METEOR:0.21, Intent: 0, Eval: 0.67
- **All-PPO:** `What is the most important step of your quest?  <EOU> Making sure that your quest succeeds`   METEOR:0.20, Intent: 0, Eval: 0.63

- $\xi$**-PPO:**     `What is your condition?  Do you have any fever?` `<EOU> No`   METEOR:0.20, Intent: 0, Eval: 0.70

- **NM-PPO:**  `I don't feel good.  I really dont feel good at all.` `<EOU> No way`   METEOR:0.21, Intent: 1, Eval: 0.66

