# OpenReview forum: "Confronting Reward Model Overoptimization with Constrained RLHF"
_ICLR.cc/2024/Conference — ICLR 2024 spotlight_

### Official Review · Reviewer_FHce · 2023-10-31

**Soundness:** 3 good
**Presentation:** 3 good
**Contribution:** 3 good
**Rating:** 6
**Confidence:** 4

**Summary:**

This paper shows that over-optimizing reward models can lead to decreased performance of LLMs. The authors remedy this through constrained reinforcement learning, where the LLM is prevented from exceeding each reward model's usefulness threshold.

**Strengths:**

- The authors address the reward model over-optimization problem with a simple constrained optimization solution.
- The authors identify that the interaction between component RMs can influence the proxy points of the component rewards.
- The authors experiment with a number of different constrained optimization solutions (e.g. $\xi$-PPO, $\mu$-PPO, etc.)
- The authors present NM-PPO to mitigate the need for multiple runs to identify proxy points.

**Weaknesses:**

- The method requires multiple runs to identify proxy points for each reward model. This is somewhat addressed by NM-PPO, but NM-PPO still requires some queries to the evaluation metric during training.

**Questions:**

- I'm curious if it's necessary in practice to satisfy all constraints. For instance, I wonder how performance would be affected if only the intent constraint was satisfied, or if only the METEOR constraint was satisfied (while still identifying proxy points through considering all RMs together)
- I'm a bit confused on why $\mu$-PPO still performs well even though the proxy point thresholds are exceeded. I do see that the authors acknowledge this as a reason why $\mu$-PPO may perform worse than $\xi$-PPO, but I wonder why $\mu$-PPO still performs well. I'm especially confused about this because in Figure 5.3 the authors show that using thresholds that are higher than the proxy point thresholds leads to much worse performance for $\xi$-PPO. Can the authors provide an explanation for this?

---

> ### Author Response · Authors · 2023-11-18
> **Authors' Response**
>
> Thank you very much for your careful review and feedback! We are glad that you appreciated the simplicity of our approach, the identification of the importance of the interaction among component RMs, the breadth of our experimentation, and the efficiency of NM-PPO. We hope to address your concerns below:
>
> Weaknesses
> - Multiple runs + Ground Truth: We absolutely agree, as we note in the Discussion section of the paper. However, we stress that the need for some access to ground truth evaluation is required of all methods, including standard PPO, as it needs multiple runs to tune the fixed weights used on the component RMs. In this sense, we can interpret the results as follows: $\xi$-PPO (and $\mu$-PPO) improve performance relative to PPO for the same number of queries to the evaluation metric. NM-PPO also improves performance relative to PPO, although to a lesser degree than $\xi/\mu$-PPO, while dramatically reducing the number of required queries to the evaluation metric as well as the number of overall training steps. Detecting overoptimization without any queries to an evaluation metric would be a significant advance for any RLHF approach, but we believe it is orthogonal to our method and outside the scope of this paper.
>
>
>
> Questions
> - Constraint Satisfaction: The answer to this depends to a large extent on the definition of “necessary.” Our results indicate that the best performance is obtained when all constraints are satisfied. If the true evaluation performance is a smooth function of the component RM scores, then a subset of component RMs deviating slightly from their constraints would likely not produce a significant drop in performance. This is what appears to have occurred with $\mu$-PPO in our case (although it should be stressed that because it is using inequality constraints, it’s not violating its constraints by exceeding the intent threshold). However, this is hard to predict and would likely vary greatly across problems.
> - $\mu$-PPO: This is likely related to the issue you raise in the previous question (and our response). $\mu$-PPO doesn’t dramatically exceed the Intent threshold, and it matches the METEOR threshold fairly accurately. In this problem, the predicted evaluation surface (Fig 3.2) is rather smooth, so this likely explains why its behavior doesn’t produce a dramatic drop in performance.
>
> Thank you very much once again for your time and consideration. We hope that we’ve managed to address your questions and concerns. If not, please let us know and we’d be more than happy to continue our discussion!

---

> > ### Comment · Reviewer_FHce · 2023-11-23
> >
> > Thank you for the discussion and for answering my questions! I will keep my score as is.

---

> ### Author Response · Authors · 2023-11-23
> **Thank you!**
>
> Thank you very much for your response and for helping to improve the paper!

---

### Official Review · Reviewer_xdFb · 2023-11-01

**Soundness:** 4 excellent
**Presentation:** 3 good
**Contribution:** 4 excellent
**Rating:** 8
**Confidence:** 3

**Summary:**

In this work, the authors improve the ability to optimize LLMs so that they are aligned with human preferences, but not overly optimized to a specific reward model (inferred via reinforcement learning from human feedback or some oracle model) or set of reward models. In this work, the authors identify the points of overoptimization, termed proxy points, and use constrained optimization to ensure that each reward model reaches, but does not surpass the proxy point. The authors explore two different approaches to determining proxy points and display that the proposed technique can avoid overoptimization and improve performance. The authors present detail regarding related work, their approach for determining proxy points, the proposed method, and a thorough evaluation section.

**Strengths:**

+ This paper is extremely well-written and contains sufficient related work and explanation to understand the proposed techniques and prior work.
+  To the best of my knowledge, the proposed technique to avoid LLM overoptimization toward composite RMs is novel.
+ The analysis is very detailed and showcases the proposed technique works as expected
+ As LLMs are a very popular topic now and actively being deployed, this approach tackles an important issue in optimizing and improving LLM performance toward high-performance behavior.

**Weaknesses:**

- Could you comment on how easy it is to identify the proxy point? In larger or multi-topic datasets, would determining a proxy point be more difficult?
- While the paper does contain much information, a lot of important information is in the appendix that would benefit from also appearing briefly in the paper. On that thread, it would be beneficial to highlight important details in D.2. Is there a specific feature of the sample outputs you are attempting to highlight?

**Questions:**

- Could you comment on how the results may differ for newer, more high-performing LLMs (e.g., GPT-4)?

- Could you also address the questions in the weaknesses above?

---

> ### Author Response · Authors · 2023-11-18
> **Authors' Response**
>
> Thank you very much for your detailed review! We’re glad you liked the paper and found it to be well-written, novel, containing thorough analysis, and of importance to the field. We hope to address your concerns below:
>
> Weaknesses
> - Proxy Point Identification: The difficulty of finding proxy points is likely more a function of the number of component RMs and their interaction with each other rather than dataset or model size. The first method in the paper (find proxy points, then optimize) is simple but expensive due to the number of runs required. (Regular PPO is also expensive in this way, because we need to search for the best fixed weighting on RMs.) NM-PPO makes this process much more efficient, but since it’s a local hill-climbing algorithm, there’s no guarantee that it’ll reach the globally optimal set of proxy points. We think improving this is an exciting direction for future work!
> - Paper Length and Sample Outputs: We absolutely agree regarding the length of the paper! Is there anything in particular that’s in the appendix that you would like to see in the main text?  Regarding the sample outputs, we were simply hoping to highlight the fact that high METEOR and intent scores do not necessarily correlate with high evaluation scores, a sign of overoptimization. We have added outputs from NM-PPO, as well as the component RM and evaluation ratings for each response to emphasize this.
>
>
>
> Questions
> - Transferability of Results: We don’t know exactly how GPT-4 was finetuned, but in principle this method should be effective for any RLHF approach. Methods like NM-PPO which eliminate the need for multiple runs and reduce the overall number of queries to the ground truth evaluation metric would be especially helpful at that scale.
> - Weaknesses: We hope we have!
>
> Thank you very much once again, and we hope we have adequately addressed your questions and concerns!

---

> > ### Comment · Reviewer_xdFb · 2023-11-21
> > **Rebuttal Response**
> >
> > Thank you for your response! Your responses have addressed my questions and concerns.

---

> > > ### Author Response · Authors · 2023-11-21
> > > **Authors' Response**
> > >
> > > Thank you, we appreciate your help in improving the paper!

---

### Official Review · Reviewer_kPJr · 2023-11-01

**Soundness:** 3 good
**Presentation:** 2 fair
**Contribution:** 3 good
**Rating:** 6
**Confidence:** 4

**Summary:**

This paper provides a novel analysis of overoptimization in the context of RLHF with multiple proxy reward functions, identifying their termed "proxy points" of overoptimization. The proxy point is a threshold at which further increasing the proxy reward results in decreased ground-truth performance. Their primary contribution is to introduce constrained RL approaches that incorporate these points into optimization objectives. Additionally, they demonstrate how a derivative-free optimization method can be used to dynamically identify the proxy points in a single run.

**Strengths:**

**Originality**

The paper presents a unique framework that introduces the concept of "proxy points" to address overoptimization in an environment with multiple proxy reward models and a given ground-truth reward model. This novel idea of defining a threshold for proxy rewards, ensuring that they don't exceed the proxy point, is a commendable original contribution to the field of RLHF.

**Quality**

The authors have validated the effectiveness of the proposed approach through empirical experiments. The application of a local hill-climbing algorithm, the Nelder-Mead threshold search, is also shown and validated through numerical experiments.


**Significance**

In the broader perspective of RLHF, managing overoptimization is a critical challenge. By estimating the thresholds that may cause a drop in performance and incorporating them in the optimization process, this paper makes a significant stride in improving the robustness and reliability of reinforcement learning models.

**Weaknesses:**

- One significant limitation, as acknowledged by the authors themselves, is the assumption of the availability of the ground-truth reward model in RLHF. While such an assumption of the gold reward model aids in understanding and analyzing overoptimization as in Gao+ 2022, its actual use within RLHF algorithms seems impractical.

- The evaluation metric described in Section A.2 is aimed at respecting both the METEOR and intent reward functions in light of Goodhart's Law. While the authors have meticulously chosen metrics from two categories - lexical quality and text diversity, there seems to be a lack of in-depth discussion about its relevance and applicability in the context of LLM-based RLHF. Drawing insights from works such as the one by Maynez et al. (Benchmarking Large Language Model Capabilities for Conditional Generation, ACL 2023) could have provided a clearer perspective on how well the proposed metric aligns with the requirements and challenges of LLM.

- The lack of detailed descriptions and elaborations of the specific methodologies presented in Table 1 raises concerns about the paper's clarity. One might find it challenging to grasp the proposed approach without sufficient explanations.

**Questions:**

- On the Validity of Evaluation Metric (Equation A.1):
  - How did the authors determine that the evaluation metric chosen in Equation A.1 accurately represents the efficacy of the proposed method?
  - Were there other potential metrics considered? If so, could the authors elucidate the rationale behind the chosen metric over others?
  - Are there any design principles or guidelines followed when setting this metric? The significance of this setting in the proposed method seems critical, and a clearer explanation would be beneficial.

- Regarding mixed advantages right after Equation (4.3): Shouldn't the term v_µ(s) encompass the second term from Equation (4.3)?

- I understand that PPO-SAT adheres to the Lagrangian multipliers method for equality constraints and thus the weights of RMs will be adjusted based on how well the constraints are met. However, in Section 5, the authors mentioned that the RM weightings in PPO-SAT are fixed. Can you please provide clarity on the workings of PPO-SAT?

- In Figure D.2, for All-PPO, why is the tanh function used for constraints that seem to be inequality constraints? This appears to be inconsistent with the explanation provided in Section 4 of the main text.

---

> ### Author Response · Authors · 2023-11-18
> **Authors' Response**
>
> Thank you very much for your careful review and helpful feedback! We are glad that you found the paper to be an original contribution, of high quality, and a significant stride towards improving RLHF. We aim to address your concerns below:
>
> Weaknesses
> - Querying the Evaluation Metric: We absolutely agree that the need to query a ground truth model is a weakness. However, as we note in the paper, this requirement is common to all the methods considered, including PPO (which requires access to the evaluation metric to tune the RM weightings). There’s currently no way to determine if overoptimization is happening without some access to ground truth evaluation. One possible route would be to make use of AI feedback (RLAIF). Such an advance would be orthogonal to the aim of this paper (and would benefit both our approach and standard PPO), and we believe it is outside the scope of this work.
> - Contextualizing the Evaluation Metric: We thank the reviewer for the reference. We found the paper insightful and have worked it into Section A.2 with a more in-depth discussion. We'd like to emphasize that the main desideratum for our evaluation metric was that it could be overoptimized in the RLHF process. In the context of the findings of Maynez et al., we note our work differs in considering RLHF on dialogue tasks, rather than open-ended natural language generation. Due to the structured nature of the task, models rarely break from the format (see generation examples in Appendix D), which permits the use of overlap metrics. To ensure relevance to the ground truth outputs, we used ROUGE2 (like Maynez et al.) in combination with some other metrics. The main difference in our evaluation metrics are the diversity metrics, which were chosen since degeneration is a common problem when finetuning language models [1]. We are excited by future extensions of our work to larger models, and more sophisticated tasks (like summarization), which would necessitate evaluations like those suggested by Maynez, et al.
> - Detailing Methodologies: Thank you for this observation! Algorithm 1 in the paper appendix comprises a detailed procedure which can be used to implement $\mu$-PPO and $\xi$-PPO, and Algorithms 2 and 3 describe the overall two-phase procedure for identifying and leveraging proxy points and NM-PPO, respectively. In addition, we included the code implementing these algorithms in the supplementary material. In the revision, we have added additional references to the algorithm descriptions in the main text as well as a description of PPO-SAT in Algorithm 2. We hope this improves the clarity of the proposed approaches!
>
> Questions
> - Evaluation Metric: Thank you for this question! As noted above, the evaluation metric was selected because it was overoptimized (i.e., obeyed Goodhart’s Law) with respect to each of the component RMs. That is, we wanted an evaluation metric that would initially increase as METEOR and intent scores increased, but would begin to decrease after a certain point, much as human evaluation ratings initially correlate with RM scores and then decline. To design this metric, we tested a number of combinations of other metrics (those included in Figure D.1), and found that the combination described in Appendix A.2 most closely adhered to the pattern of overoptimization with respect to METEOR and intent. That is, the evaluation metric was not directly chosen to simulate human preferences, but rather the way human ratings change as component RMs increase.
> - Value Function: The value expression is correct, as it measures the expected, discounted, cumulative reward with respect to the Lagrange multipliers. Furthermore,    $\sum_{i=1}^N \mu_i \theta_i$ is not a (directly) policy-dependent term.
> - PPO-SAT: We apologize for the misunderstanding! PPO-SAT is not a Lagrangian method, it sets the reward signal equal to the weighted sum of the squared errors of each RM from its corresponding proxy point, and the weights on each of these errors is fixed. It can be seen as standard PPO with a modified reward function that takes into account proxy points. We have added Algorithm 2 to the appendix, which describes this procedure in greater detail.
> - Figure D2: Thank you for catching this mistake! We mistakenly applied the wrong transformation to the saved Lagrange multiplier data, but the result is consistent. We have updated the plot in the revision accordingly and added results from $\mu$-PPO and $\xi$-PPO for comparison.
>
> Thank you very much once again for your feedback and review! We hope that the changes made and responses above have addressed your concerns. If that’s not the case, we’re more than happy to continue discussing.
>
> [1] Khalifa, M., Elsahar, H., & Dymetman, M. (2020, October). A Distributional Approach to Controlled Text Generation. In International Conference on Learning Representations.

---

> > ### Comment · Reviewer_kPJr · 2023-11-23
> >
> > Thank you for the revisions and detailed responses to the review comments. The responses have largely resolved my questions and concerns, and the revisions have notably improved the paper.
> >
> > However, I remain concerned about the high reliance on ground truth evaluation for proxy point estimation in the RLHF framework. While some dependence on ground truth is understandable, the extent of this reliance in your method may limit its practicality in scenarios where true ground truth evaluations are challenging to obtain.
> >
> > Nonetheless, the introduction of "proxy points" is a valuable advancement in RLHF. Considering the overall contributions and improvements in the revised paper, I will raise my score by one level.

---

> ### Author Response · Authors · 2023-11-23
> **Thank you!**
>
> Thank you very much! We fully agree that reducing dependence on costly evaluations is important for RLHF, and hope to address this problem further in future work. To further study this dependence in our setting, we added an additional plot (Figure D.7) which examines the effect of the number of queries per run (in the non-Nelder Mead-based approach) on the predicted evaluation surface. We appreciate your time and help in making the paper better.

---

### Official Review · Reviewer_uRgS · 2023-11-01

**Soundness:** 3 good
**Presentation:** 3 good
**Contribution:** 3 good
**Rating:** 8
**Confidence:** 3

**Summary:**

The authors analyze the problem of reward model over parameterization in the context of composite reward functions. They propose a method of identifying the points of over parameterization or proxy points taking into account the correlations between reward models. Once these proxy points are determined, the authors propose several constrained RL approaches which incorporate these points into the optimization objective.

**Strengths:**

1. The analysis of the over parameterization of composite reward functions in interesting and of importance to LLM-Alignment
2. The proposed derivative free optimization method, NM-PPO is novel and computationally efficient.
3. The paper is generally well written and easy to follow.

**Weaknesses:**

**Determining the joint maximizing point seems heuristic**

To determine the joint maximizing proxy point, the evaluation scores as a function of the METEOR and intent rewards for each run shown in Fig. 3.1 are plotted and a surface is fit over them. But these evaluations were done by maximizing each reward individually, without account of any interaction. Thus using these to determine the joint maximizing point seems heuristic at best.

In other words, if the ultimate objective is to determine the weighting among RMs to find the combination which produces the best correlation with ground truth evaluation, and to fine-tune the language model until it reaches  the proxy points of this composite RM, the proxy points obtained using the method outlined in the paper may not align with this desired outcome.

**Questions:**

1. **Proxy Points**
    1. How and how many times is the ground truth evaluation function queried to determine the maximizing proxy point? (For results in Fig 3.2 and 5.1)
    2. How is line 9 in algorithm 2 computed?
    3. Why is KL regularization not used to determine the proxy points? (results in Fig 3.1)
2. In Fig 5.1 (right) the performance of $\xi-$PPO drops after about 180k training steps, are the constraints still satisfied during this stage?
3. **Minor Points**
    1. It would be clear if you could plot $\xi-$PPO and $\mu-$PPO in figure D2 for reference. It will help in understanding the instability of All-PPO.
    2. Does All-PPO have an KL constraint?
    3. It would be helpful if  you could plot the training curve for  NM-PPO? It would be helpful to see how the changing threshold affects the training process.
    4. Sample outputs of NM-PPO are missing in section D.2
    5. Could you explain the sample outputs in section D2? its hard to understand the performance of the different algorithms. It would be helpful if you could include the evaluation score and the rewards obtained as well.

---

> ### Author Response · Authors · 2023-11-18
> **Authors' Response**
>
> Thank you very much for your detailed and helpful review! We’re glad that you appreciated our analysis of composite reward functions, the efficiency and novelty of NM-PPO, and found the paper to be well-written. We hope to address your concerns below:
>
> Weaknesses
> - Joint Proxy Point Identification: The idea behind our approach was to increase each individual RM and observe the corresponding change in the others as a way to measure their interaction. If the agent were maximizing both component RMs, then a change in one couldn’t be ascribed purely to the change in the other. Additionally, the fact that NM-PPO appears to hill-climb along the topography of the predicted surface (Fig. 5.4, and Fig. D.3) can be seen as empirical validation that our approach is in fact effective at approximating the evaluation surface. Finally, given that (to our knowledge) overoptimization has not been studied in this setting before, we believe that simple yet effective heuristics offer a useful starting place for future work, both to understand their efficacy and to design improvements. We hope to continue working on this topic, as well as inspire broader follow-up work!
>
>
> Questions
> - Proxy Points: The evaluation function is queried 20x per run in Figure 3.1. Figure 3.2 is generated using the results from the runs in Figure 3.1 (no additional queries). The algorithms do not make additional queries to the evaluation function for the results in Figure 5.1. We tested two methods for finding the maximum of the fitted evaluation function surface. First, we used the built-in implementation of L-BFGS-B in the SciPy optimization package using the results from KDE to provide bounds. Second, since the polynomial surface is approximated using a discrete mesh, we also simply picked the coordinates corresponding to the plotted maximum. There was a negligible difference between methods, so we used the second in practice. We have added this detail to the appendix. We do not use KL regularization to determine the proxy points because our goal is to isolate the influence of each component RM on the others, and KL regularization would be an additional factor influencing behavior.
> - Long Training Performance: This is a good question! Upon investigation, we found that all runs managed to continually satisfy the intent reward constraint across training, but that a single run suffered a loss of stability with respect to the METEOR reward after roughly twice the baseline training time (stability was maintained in the other seeds). This aligned with a drop in evaluation performance. We added a figure showing this, Figure D.3.
>
> Minor points:
> - Stability: We agree! We’ve added these results to the figure.
> - All-PPO: Yes, it has the same KL regularization mechanism as standard PPO. We have added text clarifying this.
> - NM-PPO Curve: We agree, and have added this in Figure D.5.
> - NM-PPO Sample Outputs: Good catch, we have added this.
> - Sample Output Rationale: We have added the reward and evaluation scores for each output. Our main point of emphasis was that high METEOR and intent scores do not necessarily correlate with high evaluation scores. Because the evaluation score was selected because it can be overoptimized by METEOR and intent, rather than any specific suitability as a measure of text quality, higher evaluation scores should not be interpreted as such. We have added text to clarify this.
>
> Thank you once again for your detailed review and comments! We hope our responses and the changes made address your concerns. If not, we’re more than happy to continue the discussion!

---

> > ### Comment · Reviewer_uRgS · 2023-11-20
> > **Rebuttal response**
> >
> > I applaud the authors for their extensive rebuttal and the modification of their paper to include additional plots and clarifications. I have modified my score accordingly.

---

> > > ### Author Response · Authors · 2023-11-20
> > > **Thank You!**
> > >
> > > Thank you very much, and thank you for your help in improving the paper!

---

### Author Response · Authors · 2023-11-18
**General Response to Reviewers**

We’d like to thank all of the reviewers for their time, as well as for their detailed reviews and feedback! We are very glad that all reviewers found our approach to be novel, the problem and results to be significant, and our analysis to be thorough. We have uploaded a revision of the paper which contains further analysis, corrects several typos, and adds more implementational details for clarity. Specifically, we have made the following additions:

- additional results to Figure D.2 depicting example optimization trajectories of $\xi$-PPO and $\mu$-PPO
- new results showing constraint satisfaction over long training runs in Figure D.3
- example training curves for NM-PPO in Figure D.5
- new Algorithm 2 describing PPO-SAT
- additional references and algorithmic details throughout the paper
- details regarding the computation of the maximum of the fitted evaluation function surface and the selection and contextualization of the evaluation metric in Appendices A.2 and A.3
- added NM-PPO and reward and evaluation scores to the sample outputs, as well as more detail on their use
- emphasized that all methods (including unconstrained baselines) require some access to ground truth evaluation to test for overoptimization

We have highlighted all changes in the paper in purple.
We address individual reviewer concerns below. Thank you all once again!

---

> ### Author Response · Authors · 2023-11-23
> **Additional Revision**
>
> We have uploaded a new revision of the paper with several additions:
>
> - To evaluate whether NM-PPO is able to scale effectively to more than two component RMs, we added BLEU score as a third RM and compared the performance to PPO. Both NM-PPO and PPO (with tuned coefficients) were run for 3 seeds. Our results show that NM-PPO is again able to outperform PPO while using 1/3 the number of total training steps. This result is shown in Figure D.2.
> - Because of the additional RM in the above result, we allowed PPO five additional tuning runs (15 total) to adjust the coefficients on the component RMs. These runs are shown in Figure D.9.
> - To visualize the effect of the number of queries to the evaluation function per run on the estimate of the evaluation function surface, we created contour plots of the predicted surface for three different query frequencies: 20 (the default), 10, and 5. This was done using the same data used to generate Figures 3.1 and 3.2. In Figure D.7, we can see that the basic structure of the predicted surface is preserved even down to 5 queries/run, indicating that much of the performance can be preserved with even more limited access to the evaluation function.
>
> We'd like to thank all the reviewers once again for their time and help in reviewing the paper!

---

### Meta-Review · Area_Chair_s6Vr · 2023-12-06

**Metareview:**

This paper introduces an approach to solve this issue using constrained reinforcement learning as a means of preventing the agent from exceeding each RM's threshold of usefulness.

**Reviewers have reported the following strengths:**

- The paper addresses an important problem;
- The proposed method is novel and interesting;
- The paper is well-written.

**Reviewers have reported the following weaknesses:**

- Assumption of the availability of the ground-truth reward model;
- Presence of many details in the appendix.

**Decision**

The paper initially received a positive assessment from the majority of Reviewers. The rebuttal phase helped further convince the Reviewers, and some of them increased the score. The paper is good for acceptance.

**Justification For Why Not Higher Score:**

The paper is a solid contribution that received a unanimous positive assessment. Although I consider this work to be good, I think it is quite technical and not interesting to a very broad audience. Thus, I consider a spotlight more appropriate than an oral presentation.

**Justification For Why Not Lower Score:**

N/A

---

### Decision · Program_Chairs · 2024-01-16

Accept (spotlight)